# ResCast: Enhancing Global Medium-Range Precipitation Forecasting with Residual Diffusion Model

## Abstract

Machine learning techniques have been successfully applied to global weather forecasting, achieving significant results across various applications. However, existing data-driven machine learning methods struggle to provide accurate medium-range meteorological predictions. As a result, precipitation forecasts regressed from these predictions are less accurate, making reliable medium-range precipitation forecasting difficult. The root causes of these issues are error accumulation in meteorological variable forecasts and a lack of effective variable interaction in the precipitation regression module. In this paper, we propose ResCast, a novel approach to global medium-range precipitation forecasting by combining meteorological residual diffusion modeling and precipitation regression. The diffusion component consists of (i) the Details Network (DetNet), which captures global features, and (ii) the Multi-Attention U-Net (AttUnet), which generates residuals for meteorological variables to reduce prediction bias. Then, a precipitation regression module quantifies the influence of residual-enhanced meteorological variables on precipitation, improving forecast accuracy. We evaluate our approach on the ERA5, an established dataset from the ECMWF, using comprehensive metrics and compare global medium-range precipitation forecasts against four state-of-the-art baselines (such as ENS, GraphCast, etc.). The results demonstrate the effectiveness and superiority of the proposed framework. The code implementation can be found in `https://anonymous.4open.science/r/ResCast-78BD`.

## 1 Introduction

Medium-range precipitation forecasting, extending predictions up to 15 days, constitutes an essential component of operational weather prediction systems, playing a pivotal role in agriculture, water resource management, and disaster prevention (Ward et al., 2011; Yue et al., 2022). However, due to the inherent nonlinearity of weather systems, forecast uncertainties progressively accumulate and amplify as the lead time increases, substantially degrading the accuracy of medium-range weather predictions (Palmer, 2000). Among the fundamental atmospheric forecast variables, precipitation forecasting presents unique challenges. This is primarily attributed to the multiscale interactions involved in precipitation processes, ranging from cloud microphysics to large-scale circulation (Frank et al., 2024), encompassing complex nonlinear dynamical, water vapor transport and thermodynamic processes (Trenberth et al., 2003). Furthermore, precipitation fields exhibit significant sensitivity to initial conditions and strong localized characteristics (Argence et al., 2008). The coupling of these factors makes precipitation substantially more challenging to predict accurately compared to other meteorological variables, a characteristic particularly pronounced in medium-range forecasts.

Traditional numerical weather prediction (NWP) systems, such as Ensemble Prediction System (ENS) and High-Resolution Forecast (HRES) (Haiden et al., 2018) from the European Centre for Medium-Range Weather Forecasts (ECMWF), provide medium-range forecasts up to 10 days through physics-based simulations, but their computational costs limit operational scalability. Thus, machine learning models (Pathak et al., 2022; Hu et al., 2023; Nguyen et al., 2023) have achieved remarkable breakthroughs in medium-range forecasting of complex chaotic weather systems, owing to their superior capabilities in modeling nonlinear systems and fast inference abilities. For example, Graphcast (Lam et al., 2022) employs adaptive mesh graph neural networks to simulate multi-scale

interactions inherent in weather dynamics. Pangu-Weather (Bi et al., 2023) designs 3D-Transformers with geophysical position encoding, outperforming IFS on upper-air variables through hierarchical training strategies. However, medium-range precipitation forecasting remains a significant challenge.

Recent years have witnessed the emergence of Machine Learning (ML) approaches in weather forecasting (Rasp et al., 2020), significantly reducing the demand for computational resources and time. However, most precipitation models (Choi et al., 2021; Zhang et al., 2023; Pathak et al., 2022) are limited to short-term predictions (within 3 days). On the one hand, while the MetNet series of models (Sønderby et al., 2020; Espeholt et al., 2022; Andrychowicz et al., 2023) pioneered machine learning for precipitation forecasting, their reliance on radar and satellite data limited their prediction range. Over time, they can only achieve the precipitation forecast for 24 hours on a regional scale. On the other hand, state-of-the-art models trained on ECMWF's ERA5 (Hersbach et al., 2018) reanalysis data, such as FourCastNet (Pathak et al., 2022), GraphCast (Lam et al., 2022) and FuXi (Chen et al., 2023), have shown their ability to forecast precipitation up to 10 days ahead. Furthermore, GenCast (Price et al., 2025) employs diffusion models to generate diverse ensemble members, enabling uncertainty quantification and enhanced prediction of extreme weather events. These ML-based approaches significantly reduce computational time from hours to minutes.

Although existing data-driven machine learning models trained on ERA5 data have reached the capability for medium-range precipitation forecasting, they have yet to surpass traditional NWP models. This limitation primarily arises because mainstream precipitation forecasting methods rely on first predicting **fundamental meteorological variables** (e.g., wind, temperature, humidity, and pressure) and subsequently **regressing precipitation** from these meteorological variables (Chen et al., 2023; 2024; Lam et al., 2022; Price et al., 2025). However, prediction errors in the fundamental meteorological variables gradually increase over the 5–15 day horizon, leading to decreased accuracy in precipitation regression forecasts (Cheng et al., 2025), as shown in Fig. 1. Overall, two core challenges currently constrain the forecasting performance of these methods: (1) **Error Accumulation** in predicting fundamental meteorological variables; (2) insufficient modeling of **Variable Interaction** during precipitation regression. To achieve more accurate medium-range precipitation forecasting, we propose ResCast, which enhances the prediction of fundamental meteorological variables by integrating a residual component. The refined variables are then fed into a precipitation regression module that accounts for the influence of different predictors, thus boosting overall forecasting accuracy.

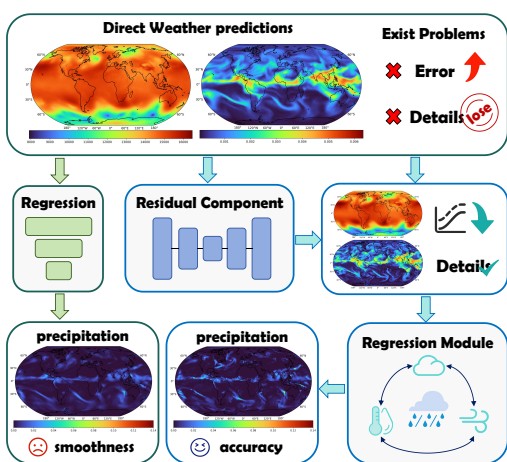

Figure 1: **Precipitation Inference diagram.** The left green arrow denotes the mainstream approach that regresses precipitation directly from meteorological variables predictions. The right blue arrow represents our proposed inference mechanism for improving precipitation regression accuracy.

In this paper, we propose a diffusion-based residual component to enhance detailed features within these variables. To construct the residual component, we designed the Details Network (DetNet) to capture the global trend information of meteorological variables, and then the global trend information is incorporated into the Multi-Attention Unet (AttUnet) as a diffusion condition to generate the residual of the meteorological variables. This approach of direct prediction combined with residual prediction mitigates error in medium-range weather forecasting. Furthermore, we model the causal relationships between atmospheric variables and precipitation using our designed precipitation regression module. Specifically, this module quantifies the important factors influencing precipitation under various meteorological variables, thus improving the accuracy of precipitation forecasting.

In summary, our main contributions are as follows:

- We design a diffusion-based residual component, including the Details Network (DetNet) and Multi-Attention Unet (AttUnet), to effectively predict residuals, reducing error for fundamental meteorological variables.

- We introduce a precipitation regression module to quantify the interactions between meteorological variables. This mechanism improves the accuracy of precipitation forecasts.

- We propose ResCast, a residual-based forecasting framework for global medium-range precipitation prediction (up to 15 days) that surpasses machine learning baselines (GraphCast, FourCastNet) and state-of-the-art traditional NWP models (ENS, HRES).

## 2 TASK FORMULATION

Following the approach in  (Pathak et al., 2022; Lam et al., 2022), our task is the problem of deterministic prediction for grid-based reanalysis data ERA5  (Hersbach et al., 2018). The process is **two stages**: (i) we generate residuals to reduce over-smoothing in fundamental meteorological prediction; (ii) these are input into a precipitation regression model to predict precipitation.

Specifically, given the meteorological variables data $x_t \in \mathbb{R}^{C \times H \times W}$ at time $t$, where $H$ and $W$ represent the spatial resolution of the frame, and $C$ is the number of fundamental meteorological variables, the meteorological predictor $\mathcal{M}_{\theta_1}$ performs a single-step inference to obtain the prediction $\hat{x}_{t+1} = \mathcal{M}_{\theta_1}(x_t) \in \mathbb{R}^{C \times H \times W}$ at time $t + 1$.

Subsequently, the residual of meteorological variables prediction $r_{t+1} \in \mathbb{R}^{C \times H \times W}$ is the residuals between the true meteorological variables $x_{t+1}$ and the predicted meteorological variables $\hat{x}_{t+1}$, and residual diffusion component ($\mathcal{M}_{\theta_2}$ & $\mathcal{M}_{\theta_3}$) generates the residuals prediction $\hat{r}_{t+1} \in \mathbb{R}^{C \times H \times W}$, formally expressed as:

$$r_{t+1} = x_{t+1} - \hat{x}_{t+1}, \quad \hat{r}_{t+1} = p_\theta \left( r_{t+1}^0 \mid r_{t+1}^K \right) \tag{1}$$

where, we use the superscript $K$ for the $K$-step reverse diffusion process. Hence, $\hat{y}_{t+1} \in \mathbb{R}^{C \times H \times W}$ denotes the final weather prediction with residual prediction $\hat{y}_{t+1} = \hat{x}_{t+1} + \hat{r}_{t+1}$.

The prediction with residuals serves as input to the precipitation regression model $\mathcal{M}_{\theta_4}$ to predict the precipitation value $\hat{tp}_{t+1} = \mathcal{M}_{\theta_4}(\hat{y}_{t+1}) \in \mathbb{R}^{1 \times H \times W}$ at time $t + 1$.

Furthermore, we selected 22 atmospheric variables (as shown in Table 9) that significantly influence precipitation in this task. More details are summarized in the Appendix E.2.

## 3 RESCAST

Fig. 2 illustrates the details of the two components of the ResCast framework: (a) a **Fundamental Meteorological Predictor with Residual Diffusion Component** generates residuals based on the original prediction, improving the error of the meteorological forecast and (b) a **Precipitation Regression Component** that uses the previous meteorological forecast as input to regress and predict precipitation. We illustrate their details as follows.

### 3.1 FUNDAMENTAL METEOROLOGICAL PREDICTOR

The Fundamental Meteorological Predictor is primarily based on the Vision Transformer (ViT) (Yuan et al., 2021), which effectively captures spatial information to perform weather image frame prediction tasks. This approach has been shown to achieve results in  (Nguyen et al., 2023).

To obtain more accurate predictions for the next time step $\hat{x}_{t+1} \in \mathbb{R}^{C \times H \times W}$, the parameters of the deterministic predictor are primarily optimized using the Lat-weighted MSE, following the approach in  (Nguyen et al., 2023; Bodnar et al., 2024). The parameters of the Fundamental Meteorological Predictor $\mathcal{M}_{\theta_1}$ are trained before the diffusion training process.

While this direct prediction effectively captures the low-frequency information of climate change  (Li et al., 2025), particularly the global motion trend, it still suffers from significant smoothing and bias in the predicted values. Hence, the parameters of this component are **updated again** for the training of the residual diffusion component.

Figure 2: **(a) Diffusion Model Training.** During this stage, we update the parameters of both the Fundamental Meteorological Predictor and the Residual Diffusion Component. The Residual Diffusion Component learns to denoise under the supervision of the true residuals. **(b) Precipitation Inference.** After the Fundamental Meteorological Predictor produces the next predictions with residuals, which are then input into the Precipitation Regression Component to regress precipitation forecasts.

## 3.2 RESIDUAL DIFFUSION COMPONENT

As shown in Fig. 2(a), we use the diffusion model framework to model the generation of global residual information, which helps address the limitations of direct prediction. The residual $r_{t+1}$ in Eq. 1 is used as supervised information during training.

**Physics-Constrained Residual Evolution** Based on the theoretical formulation in Eq. 12 in Appendix C.1, suppose we use a K-step denoising diffusion process to model the distribution:

$$p_{\theta_2}(r_{t+1}^{0:K}|\hat{r}_{t+1}) = p(r_{t+1}^K) \prod_{k=1}^{K} p_{\theta_2}(r_{t+1}^{k-1}|r_{t+1}^k, \hat{r}_{t+1}), \tag{2}$$

where $r_{t+1}^K \sim \mathcal{N}(0, I)$ and K is the number of denoising step. $\theta_2$ represents the parameters of $\mathcal{M}_{\theta_2}$. Similar to adding noise at step $k$, which allows $r_{t+1}^k$ to recover $r_{t+1}^{k-1}$. Hence, we set the noise prediction bias as the optimization objective. Given that the denoising function $\epsilon_{\theta_2}(r_{t+1}^k, \hat{r}_{t+1}, k)$ is shared across all steps of the diffusion process, we have the following objective function:

$$\mathcal{L}_\epsilon = \mathbb{E}_{(r_{t+1})\sim r, t, \epsilon \sim \mathcal{N}(0,I)} \left\| \epsilon - \epsilon_{\theta_2}(r_{t+1}^k, \hat{r}_{t+1}, k) \right\|^2 \tag{3}$$

where the $k$-th denoising state $r_{t+1}^k$ can be computed as:

$$r_{t+1}^k = \sqrt{\overline{\alpha}_k}(\underbrace{x_{t+1} - \mathcal{M}_{\theta_1}(x_t)}_{residual}) + \sqrt{1 - \overline{\alpha}_k}\epsilon. \tag{4}$$

We refer to the objective in Eq. 3 as the denoising loss. To ensure that the predicted residuals of the meteorological variables comply with physical constraints, we use the Conservation Law Loss function to enforce the conservation laws (as detailed in Appendix C.3):

$$\mathcal{L}_{\text{conservation}} = \frac{1}{N} \sum_{i=1}^{N} (x_{t+1} - \mathcal{M}_{\theta_1}(x_t))_i^2 \tag{5}$$

Finally, we use the following combined loss function to guide the residual predictions towards solutions that adhere to physical principles:

$$\mathcal{L} = \alpha \mathcal{L}_\epsilon + (1 - \alpha)\mathcal{L}_{\text{conservation}} \tag{6}$$

where $\alpha$ is a learnable weighting factor that balances the two losses.

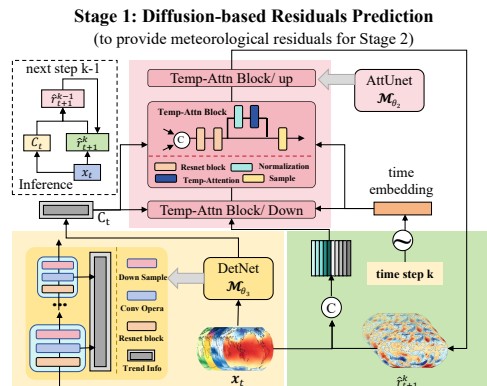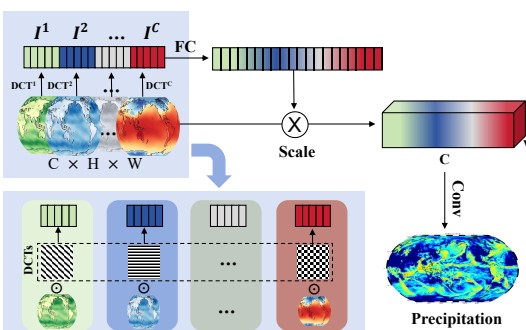

Figure 3: **Illustration of the proposed two-stage framework. Stage 1** uses the Detail-Enhancing Network, DetNet (yellow) to extract global motion information and the Multi-Attention Unet, AttUnet (red) to capture spatiotemporal evolution in a diffusion-based architecture. **Stage 2** performs precipitation regression, where DCT bases are used to regress precipitation accurately.

**Detail-Enhancing Network (DetNet)**  As shown in the yellow part of Fig. 3 (Stage 1), we designed a structure similar to ResCNN (Georgescu et al., 2020). **DetNet** $\mathcal{M}_{\theta_3}$, where $\theta_3$ represents the parameters of DetNet, is a multi-layer architecture with multiple ResBlocks. Each residual block consists of a sampling operator, a convolutional operator, and a residual network (ResNet) operator. **DetNet** can extract variable features from the current weather state, denoted as $\mathcal{C}_t = \mathcal{M}_{\theta_3}(x_t)$. We refer to this meteorological feature $\mathcal{C}_t$ as **Global Trend Knowledge**. DetNet can use convolution to extract weather motion features from the initial weather frame.

**Muti-Attention Unet (AttUnet)**  To generate more accurate fundamental meteorological variable residuals, **AttUnet** $\mathcal{M}_{\theta_2}$ is designed with spatial attention blocks, as shown in the red part of Fig. 3 (Stage 1). It is a variant of the UNet in DDPM. The Temp-Attn block combines several operators: a connection operator, a ResNet operator, a normalization operator, and a channel attention unit, followed by up/down sampling operators. It will use $\mathcal{C}_t$ as conditional information for the diffusion component:

$$\hat{r}_{t+1} = p_{\theta_2}(r_{t+1}|x_t - \mathcal{M}_{\theta_1}(x_t), \mathcal{C}_t), \tag{7}$$

and the denoiser function is $\epsilon_{\theta_2}(r_{t+1}^k, \hat{r}_{t+1}, \mathcal{C}_t, k)$.

### 3.3 Precipitation Regression Modeling

Various approaches have been proposed for the precipitation regression task based on predicting fundamental meteorological variables (Pathak et al., 2022; Lam et al., 2022; Nguyen et al., 2023). However, these models fail to dynamically quantify the important factors influencing precipitation under different meteorological variables.

As shown in the light blue part of Fig. 3 (Stage 2), we propose a Precipitation Regression Component that applies multiple frequency components of the two-dimensional Discrete Cosine Transform (2DDCT) (Ahmed et al., 2006) to calculate the influence scalar for each channel, formed as:

$$B_{h,w}^{i,j} = \cos(\frac{\pi h}{H}(i + \frac{1}{2}))\cos(\frac{\pi w}{W}(j + \frac{1}{2})). \tag{8}$$

and each channel (meteorological variable) information is compressed as $I^c \sim \mathbb{R}^C$, written as:

$$I_{t+1}^c = 2\text{DDCT}(\hat{x}_{t+1}^c + \hat{r}_{t+1}^c) = 2\text{DDCT}(\hat{y}_{t+1}^c) = \sum_{i=1}^{H}\sum_{j=1}^{W} \hat{y}_{t+1}^{c,i,j} B_{t+1}^{i,j} \tag{9}$$

where $c \in \{1, 2, \cdots, C\}$, and the entire compressed vector is obtained by concatenation: $\hat{I} = [I^0, I^1, \cdots, I^C] \in \mathbb{R}^{C \times H \times W}$. The whole precipitation regression can be written as:

$$\hat{tp} = \text{Conv}_{C \to 1}(\sigma(fc(\hat{I})) \cdot \hat{y}) \tag{10}$$

where $\text{Conv}_{c \to 1}$ represents a convolution layer that reduces the number of channels from $C$ to 1, $\sigma$ is the Sigmoid function, and $fc$ represents a fully connected layer.

Table 1: Comparison of Model Performance Across Metrics and Forecast Days.

| Metrics | Model | 5 | 6 | 7 | 8 | 9 | 10 | 11 | 12 | 13 | 14 | 15 |
|---------|-------|---|---|---|---|---|----|----|----|----|----|----|
| **ACC (↑)** | FourCastNet | 0.164 | 0.128 | 0.094 | 0.068 | 0.057 | 0.054 | - | - | - | - | - |
| | GraphCast | 0.226 | 0.218 | 0.213 | 0.214 | 0.216 | 0.214 | - | - | - | - | - |
| | HRES | 0.178 | 0.172 | 0.167 | 0.163 | 0.161 | 0.160 | - | - | - | - | - |
| | ENS | 0.312 | 0.309 | 0.305 | 0.302 | 0.298 | 0.291 | 0.281 | 0.272 | 0.261 | 0.255 | 0.251 |
| | ResCast (ours) | **0.361** | **0.351** | **0.340** | **0.328** | **0.314** | **0.307** | **0.294** | **0.284** | **0.275** | **0.268** | **0.254** |
| **RMSE (↓)** | FourCastNet | 9.200 | 9.289 | 9.398 | 9.879 | 9.975 | 10.210 | - | - | - | - | - |
| | GraphCast | 6.911 | 7.023 | 7.032 | 7.124 | 7.006 | 7.115 | - | - | - | - | - |
| | HERS | 8.032 | 8.056 | 8.167 | 8.087 | 8.073 | 8.063 | - | - | - | - | - |
| | ENS | 6.341 | 6.355 | 6.371 | 6.388 | 6.416 | 6.453 | 6.511 | 6.582 | 6.668 | 6.754 | 6.849 |
| | ResCast (ours) | **5.872** | **5.943** | **6.081** | **6.124** | **6.313** | **6.384** | **6.435** | **6.498** | **6.514** | **6.604** | **6.804** |
| **Ts@1mm (↑)** | FourCastNet | 0.241 | 0.228 | 0.218 | 0.210 | 0.205 | 0.200 | - | - | - | - | - |
| | GraphCast | 0.415 | 0.412 | 0.410 | 0.409 | 0.391 | 0.382 | - | - | - | - | - |
| | HERS | 0.384 | 0.371 | 0.367 | 0.375 | 0.349 | 0.336 | - | - | - | - | - |
| | ENS | 0.458 | 0.457 | 0.457 | 0.456 | 0.455 | 0.453 | 0.450 | 0.446 | 0.441 | 0.436 | 0.430 |
| | ResCast (ours) | **0.513** | **0.497** | **0.486** | **0.478** | **0.463** | **0.457** | **0.454** | **0.451** | **0.448** | **0.445** | **0.437** |
| **F1@1mm (↑)** | FourCastNet | 0.389 | 0.372 | 0.358 | 0.347 | 0.339 | 0.334 | - | - | - | - | - |
| | GraphCast | 0.586 | 0.574 | 0.572 | 0.562 | 0.543 | 0.538 | - | - | - | - | - |
| | HERS | 0.554 | 0.541 | 0.543 | 0.534 | 0.521 | 0.512 | - | - | - | - | - |
| | ENS | 0.628 | 0.627 | 0.627 | 0.626 | 0.625 | 0.624 | 0.621 | 0.616 | 0.612 | 0.607 | 0.601 |
| | ResCast (ours) | **0.651** | **0.644** | **0.635** | **0.629** | **0.628** | **0.627** | **0.624** | **0.622** | **0.619** | **0.614** | **0.606** |

## 4 EXPERIMENT

**Implementation details.** Following in (Nguyen et al., 2023), we choose the Vision Transformer (Dosovitskiy et al., 2020) as the lightweight weather predictor $\mathcal{M}_{\theta_1}$, due to its ability to learn spatial features. In the Diffusion Training Process, we utilized the training settings in Table 4 to train the Residual Diffusion Component (DetNet$\mathcal{M}_{\theta_2}$ & AttUnet$\mathcal{M}_{\theta_3}$) for residual generation. The Precipitation Regression Component primarily focuses on training the ability to regress precipitation from multi-variable meteorological variables (parameter setting in Table 8). We trained the model on eight GPUs with 80 GB of memory each, using the PyTorch Lightning framework (Sawarkar, 2022), and achieved a total throughput of 77.6 TFLOPS. We used the global ERA5 dataset (Hersbach et al., 2018) for our experiments, with more details in Appendix D.1.

**Metrics.** Following in (Pathak et al., 2022; Lam et al., 2022; Sønderby et al., 2020), we quantify the performance of precipitation prediction using ACC and RMSE, and then evaluate its forecasting capability through TS (same as CSI) and F1. In addition, to measure the visual quality of meteorological variable prediction, we also calculate the RMSE, LPIPS (Zhang et al., 2018), and SSIM (Wang et al., 2004). More details for these metrics can be found in Appendix E.3.

**Baselines.** For precipitation forecasting performance, the baseline models compared with our proposed model include traditional numerical weather prediction models, as HERS and ENS, as well as machine learning (ML) models like FourCastNet (Pathak et al., 2022) and GraphCast (Lam et al., 2022). Except for ENS, which has a 15-day precipitation forecast, other models can only provide 10-day forecasts. The results of these baselines are partially available for download in WeatherBench2 (Rasp et al., 2024) (more details in Appendix D.3).

### 4.1 EXPERIMENTAL RESULTS

**Precipitation Forecast Comparison** We display the model performance from day 5 to day 15 in Table 1. Based on these results, we have the following observations:

- In all metrics, our proposed ResCast consistently outperforms all baselines. Specifically, compared with the existing state-of-the-art model ENS, ResCast improves ACC by an average of 0.022, reduces RMSE by an average of 0.182, and improves Ts and F1 by an average of 0.017 and 0.008, respectively, in 5-15 day forecasts with a 1mm precipitation threshold. These results represent significant improvements.

- Based on different modeling approaches, we compared and found that early ML-based methods, such as FourCastNet, did not achieve good precipitation forecasting capabilities, with low performance across all metrics. However, the Recent ML-based model, GraphCast, outperformed the

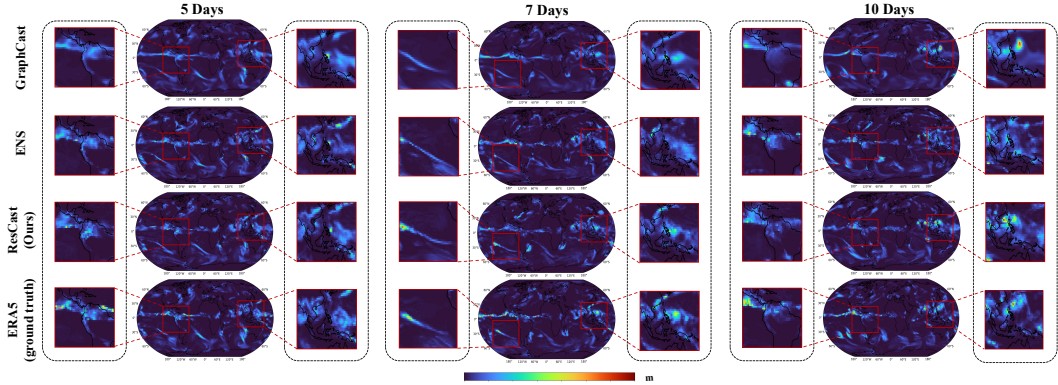

Figure 4: **Forecast Visualization**. Comparing outputs from three models (ours, ENS, GraphCast). The forecast example was initialized at 2018-05-31 00:00 UTC, with plots corresponding to 5, 7, and 10-day lead times.

numerical model HRES, but lagged behind ENS, also a numerical model. ResCast, also an ML-based model, achieved superior performance, demonstrating the significant value of this approach. Furthermore, we present our comparison results with the state-of-the-art ML-based model GenCast (see Appendix G.3 for details).

• By comparing ResCast with traditional numerical methods and recent advanced machine learning methods (such as GraphCast), the performance of ResCast highlights that machine learning-based methods, especially the strategy of using residual prediction, can significantly improve the accuracy and stability of precipitation forecasts.

**Forecast Visualization**  We provide a visual comparison of the numerical method ENS and the ML-based model GraphCast, which have shown better performance, in Fig. 4. We can find that in the 5, 7, and 10-day performance, the distribution of our local precipitation is closer to the ground truth than ENS and GraphCast. For the performance of local extreme precipitation, we conduct a local zoom comparison. Precipitation is mainly concentrated in East Asia and Central America. Even on a time scale as long as 10 days, our extreme value performance is still closer to the actual situation than the baselines, reflecting the high accuracy of our model's precipitation forecast.

### 4.2 ABLATION STUDY

**Performance of Residual Diffusion Component**  To quantitatively assess how foundational meteorological variables (e.g., wind temperature, humidity, and pressure) forecast performance changes with lead time and verify the positive impact of the Residual Diffusion Component, we employ climatology[1] and direct weather predictor (ViT) outputs with or without Residual forecasts for comparison. We plot the metrics curves for 2 metre temperature (T2m) and 10 metre wind speed (Wind10) in Fig. 5, and visualize the error changes for T2m after adding residual prediction in Fig. 6. More variable results in Appendix G.1. We can obtain the following observation:

• Whether it is climatology or the weather predictor, as the lead time increases, the performance of all methods decreases, because the uncertainty for prediction is enlarged. However, the performance of methods with the Residual Diffusion Module is always better than that without it. This again validates the effectiveness of the proposed Residual Diffusion Module.

• As the lead time increases, the errors seem to spread and intensify in certain regions of the globe. The ViT+Diff model appears to show slightly less error in many areas compared to the ViT model. Specifically, the ViT+Diff model appears to have lower error across the globe, particularly in South America, North America, and the Southern Ocean. The incorporation of residuals likely helps improve forecast accuracy in these regions, suggesting the importance of the residual-based approach for weather forecasting.

---

[1]Based on the 1990–2019 30-year average (366 days).

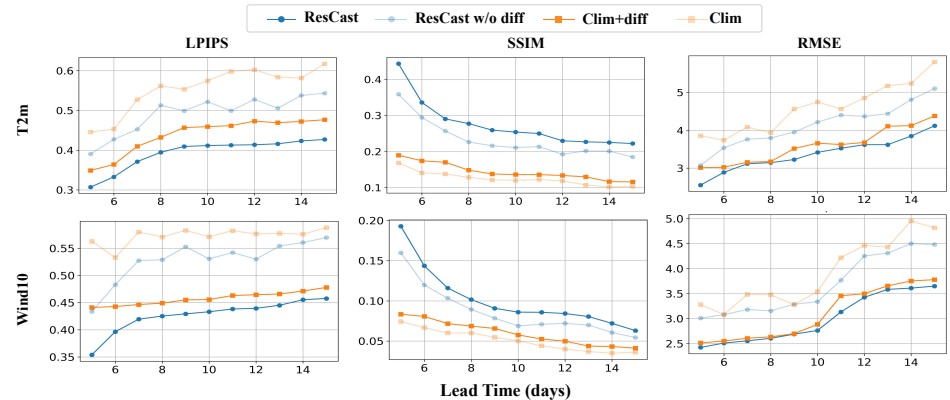

Figure 5: Performance metrics (LPIPS↓, SSIM↑, RMSE↓) across lead times for T2m and Wind10.

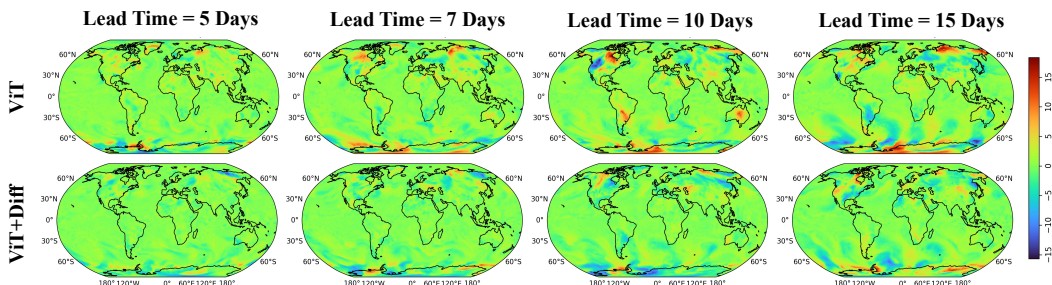

Figure 6: Visual examples of the error changes for T2m after adding residual prediction (+Diff). Example initialized at 2018-05-31 00:00 UTC, corresponding to 5, 7, 10, and 15-day lead times.

**Effectiveness of DetNet $\mathcal{M}_{\theta_3}$ and Regression Component.** We show the ablation study on DetNet in Table 2 that removing the DetNet significantly degrades performance for Q700 and T2m, which indicates its effectiveness. Furthermore, compared with standard regression models (U-Net, ViT, AFNO), our precipitation regression module achieves improvements of $5\% \sim 10\%$ across multiple metrics, demonstrating superior performance.

**Impact of Conservation Law Loss.** In the diffusion training process, our framework has two loss functions: the denoising loss and the conservation law loss, shown in Eq. 6. The conservation law loss warrants further investigation regarding its impact. As shown in Fig. 7, we evaluate the Energy Conservation Metric (ECM), defined as the ratio of predicted to real total atmospheric energy. Incorporating consistently produces ECM values that are closer to 1 compared to those without it, indicating that predictions more closely adhere to the physical principles of energy conservation.

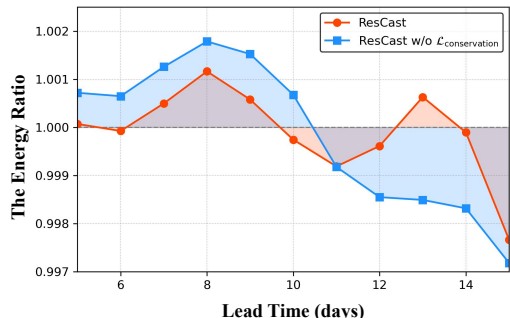

Figure 7: Analysis of Energy Ratio over time.

## 5 RELATIVE WORK

### 5.1 PRECIPITATION FORECASTING WITH MACHINE LEARNING

Recent advances in precipitation forecasting with machine learning are summarized as two paradigms: accurate deterministic modeling and probabilistic approaches for uncertainty quantification.

**Deterministic forecasting** involves predicting atmospheric variables with a single, most likely outcome. In the deterministic forecasting models, while the MetNet series (Sønderby et al., 2020; Espeholt et al., 2022; Andrychowicz et al., 2023) and DiffCast (Yu et al., 2024) have proposed accurate

Table 2: Ablation of ResCast wrt DetNet $\mathcal{M}_{\theta_3}$(left). Analysis of Precipitation Regression Module for Precipitation Prediction Using ERA5(right).

| Model | Q700 | | | T2m | | | Model | tp | | |
|---|---|---|---|---|---|---|---|---|---|---|
| | RMSE (1e-3) | LPIPS | SSIM | RMSE | LPIPS | SSIM | | ACC | Ts@1mm | F1@1mm |
| Clim | 1.782 | 0.307 | 0.111 | 3.012 | 0.367 | 0.167 | U-Net | 0.612 | 0.493 | 0.580 |
| Clim+Diff | 1.535 | 0.259 | 0.133 | 3.155 | 0.354 | 0.189 | AFNO | 0.646 | 0.514 | 0.610 |
| ResCast w/o $\mathcal{M}_{\theta_3}$ | 1.674 | 0.225 | 0.233 | 2.889 | 0.346 | 0.259 | ViT | 0.589 | 0.472 | 0.540 |
| ResCast | **1.332** | **0.199** | **0.267** | **2.551** | **0.307** | **0.277** | Ours | **0.652** | **0.531** | **0.650** |

machine learning models for short-term precipitation forecasting, their reliance on radar and satellite data limits the forecast horizon to 24 hours. Hence, achieving reliable medium-range precipitation forecasts requires first predicting meteorological variables, such as wind, temperature, humidity, and pressure, before regressing precipitation (Jennings et al., 2025). For example, FourCastNet (Pathak et al., 2022) and GraphCast (Lam et al., 2022) use neural networks suited for global motion modeling to forecast atmospheric variables and precipitation regression. Meanwhile, FuXi (Chen et al., 2023) proposes a multi-model cascaded approach for medium-range precipitation prediction. However, these models' self-regressive iterative forecasts lead to error accumulation after 5 days, affecting the accuracy of precipitation regression.

**Probabilistic forecasting** has traditionally relied on ensemble prediction systems such as ECMWF-ENS (for Medium-Range Weather Forecasts, 2024), where forecast uncertainties are sampled through ensemble members with varied initial conditions. GenCast (Price et al., 2025) innovatively applies diffusion models to generate ensemble members through denoising different initialization noises to achieve medium-range probabilistic forecasting. This approach has demonstrated superior skill compared to traditional ensemble prediction systems. However, global diffusion processes often directly generate precipitation without considering the influence of individual meteorological variables on precipitation, violating physical constraints (e.g., energy conservation). Additionally, they either lack stability or ignore forecasting error propagation.

### 5.2 DIFFUSION MODELS FOR DETAILS GENERATION

Diffusion models capture uncertainty through iterative denoising from noise to data. In weather forecasting, the coupled dynamics between initial condition sensitivity and nonlinear physical processes induces inherent uncertainties and chaotic behavior (Palmer, 2019). This natural capability of diffusion models to handle stochastic processes makes them particularly well-suited for weather forecast (Li et al., 2024). Recent applications have demonstrated promising results in various weather prediction tasks (Andrae et al., 2024; Nai et al., 2024; Shi et al., 2024).

Compared to probabilistic forecasting (Price et al., 2025), deterministic diffusion-based methods enhance spatial-temporal dependency modeling via conditional control mechanisms (Voleti et al., 2022; Martin & Šimánek, 2024). This is crucial for medium-range forecasting, where temporal smoothing occurs, as these models restore localized meteorological features (Zhao et al., 2024). However, direct precipitation generation via diffusion introduces randomness, reducing the local accuracy of precipitation distributions and failing to meet medium-range forecasting needs.

### 6 CONCLUSION

In this paper, we propose ResCast, a novel framework for global medium-range precipitation forecasting through a two-step combined forecast. We proposed a residual diffusion component and a precipitation regression module. The residual diffusion component generate residuals prediction to reduce errors in fundamental meteorological variable predictions. The precipitation regression module then quantifies the influence of these variables to improve precipitation regression accuracy. Through comprehensive experiments on ERA5 data, our proposed framework demonstrates superior accuracy and stability, consistently outperforming both machine learning baselines and traditional NWP models. In future work, we will further explore the impact of ensemble forecasting on precipitation tasks and improve the forecasting capability of the model. We will try to combine data from more sources, such as more reliable and effective data such as satellites and radars, and complete at least one case study of extreme precipitation to achieve model reliability.

## 7 REPRODUCIBILITY STATEMENT

We are committed to ensuring the reproducibility of our research. Source code and a README.md file with detailed instructions for data preparation and script execution are available at `https://anonymous.4open.science/r/ResCast-78BD` and also provided in the Appendix. The appendix offers comprehensive details to support our claims. Appendix C supplements the derivation of the formula principle of our residual diffusion module, the reasoning flowchart of our algorithm, and the algorithm implementation details. Furthermore, Appendix D lists in detail the sources and processing methods of the experimental data. Appendix E describes the details and hyperparameters of our proposed ResCast and baselines, and the calculation formulas of the metrics are also included.

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

# ResCast: Enhancing Global Medium-Range Precipitation Forecasting with Residual Diffusion Model

## (APPENDIX)

TABLE OF CONTENTS IN APPENDIX

# A    UTILIZATION OF LARGE LANGUAGE MODELS (LLMs)

In this research, Large Language Models (LLMs) were employed as a supportive tool to improve the overall quality and clarity of the manuscript. While the core research, analysis, and intellectual contributions are entirely the work of the authors, LLMs played a crucial role in refining and polishing the text. The specific ways in which LLMs were applied are as follows:

- **Grammar and Style Enhancement:** The LLMs assisted in ensuring grammatical accuracy and improving the sentence structure throughout the manuscript. They also helped in maintaining consistent phrasing and tone, which elevated the readability of the paper.

- **Improving Logical Flow:** LLMs were utilized to help organize and clarify the logical progression of ideas. By suggesting improvements in paragraph transitions, they contributed to a more cohesive and fluid narrative.

- **Coherence and Structure:** LLMs were used to organize and strengthen the logical flow of ideas. Suggestions were incorporated to improve transitions between sections and paragraphs, making the narrative more cohesive.

Each suggestion provided by the LLM was thoroughly reviewed, edited, and approved by the authors, ensuring it accurately reflected the research goals and intent. Ultimately, the responsibility for the manuscript's content rests entirely with the authors.

# B    NOTATIONS

Table 3 summarizes the notations appearing in this paper.

Table 3: Summary of key notations.

| Symbol | Description |
|---|---|
| $K, k$ | number, index of denoising steps |
| $T, t$ | number, index of time points |
| $C, c$ | number, index of channels |
| $H, W$ | height, width of variable resolution |
| $x, \hat{x}$ | true value, prediction of the fundamental meteorological variables (without residuals) |
| $r, \hat{r}$ | true value, prediction of the residuals between the ground true and prediction of meteorological variables |
| $y, \hat{y}$ | true value, prediction of the fundamental meteorological variables with residuals |
| $tp, \hat{tp}$ | true value, prediction of the total precipitation |
| $p_\theta$ | reverse process model, parameterized by $\theta$ for generating the true residual from the predicted residual |
| $q$ | forward noise process model, adding noise to the residuals |
| $\alpha, \overline{\alpha_k}$ | $\alpha = 1 - \beta, \overline{\alpha_k} = \prod_{k=1}^{K} \alpha_k$ (coefficients used in diffusion model) |
| $\beta$ | is predefined by an incremental variance schedule |
| $\mu$ | posterior mean function of the reverse denoising process |
| $\epsilon$ | noise term added during the forward diffusion process, also the main trainable parameter for the model |
| $\theta$ | Parameters of the model, used in the denoising process and reverse function |
| $\sigma$ | a variance hyperparameter, controlling the noise variance during the process |
| $\mathcal{M}$ | model functions (e.g., model for motion information) |
| $\mathcal{L}_\theta$ | loss function, typically for training the parameters $\theta$ of model |
| $L(i)$ | the $i$-th latitude weighting factor |
| $\mathcal{C}$ | the extracted global motion information from the frame of meteorological variables |
| $B$ | basis functions for multi-level channel attention, related to the Discrete Cosine Transform (DCT) |
| $I$ | input vector, representing multi-level or multi-channel input in the model |

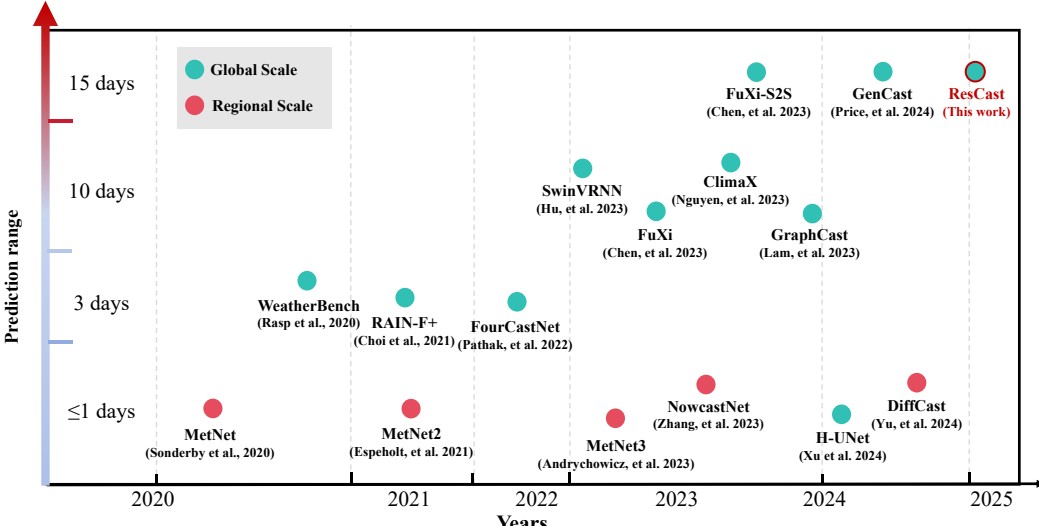

Figure 8: **Development Trends of Precipitation Models.** This figure illustrates the development trend of precipitation forecasting models. As the accuracy of global-scale precipitation models improves, there is a clear shift towards extending the forecast range, moving towards longer-term predictions. Simultaneously, there is an emphasis on optimizing models for more localized and finer-scale forecasts.

## C SUPPLEMENT OF MODELING DETAILS

### C.1 PRELIMINARY: DIFFUSION MODELS

Denoising Diffusion probabilistic models (DDPMs) (Ho et al., 2020; Song et al., 2020) aim to learn the data distribution $p(r)$ by training the model to reverse the Markov noise process that gradually degrades the data. For ease of explanation, all instances of $r$ below refer to the residuals at the same time $t + 1$.

Specifically, diffusion models first involve a forward diffusion process, which gradually adds noise to the real image in a non-parametric manner. Then, a reverse denoising process is used, where a parameterized network learns the noise generation, allowing it to denoise and generate the target state from pure Gaussian noise. Hence, we can obtain the $k$-th diffusion state from the residual climate values $r_0$, which has a special property:

$$q(r^k|r^0) = \mathcal{N}(r^k; \sqrt{\overline{\alpha}_k}r^0, (1 - \overline{\alpha}_k)I), \tag{11}$$

where $r^K \sim \mathcal{N}(0, 1)$ as a sample drawn from a pure Gaussian distribution, while $r^0$ is derived from the target residual values. The coefficient $\overline{\alpha_k} = \prod_{k=1}^{K} \alpha_k$, where $\alpha_k = 1 - \beta_k$ and $\beta_k \in (0, 1)$ is predefined by an incremental variance schedule. With this, we can infer the reverse denoising process, which is governed by a Markov chain defined by parameterized Gaussian transitions:

$$p_\theta(r^{k-1}|r^k) = \mathcal{N}(r^{k-1}; \mu_\theta(r^k, k), \sigma_k^2 I), \tag{12}$$

where $\mu_\theta$ represents the posterior mean function. The reverse function $p_\theta(r_{k-1}|r_k)$ aims to remove the noise added during the forward noise process, and the parameterized network is defined as:

$$\mu_\theta(r^k, k) = \frac{1}{\sqrt{\alpha_k}}(r^k - \frac{\beta_k}{\sqrt{1 - \overline{\alpha}_k}}\epsilon_\theta(r^k, k)), \tag{13}$$

where $\epsilon_\theta(r_k, k)$ is the main trainable parameter function of diffusion models, which estimates the corresponding noise at the current time step $k$ and is applied during the denoising process. The parameter $\theta$ can be learned using the following loss function as the objective:

$$L(\theta) = \mathbb{E}_{r^0,k,\epsilon}\|\epsilon - \epsilon_\theta(r^k, k)\|^2, \tag{14}$$

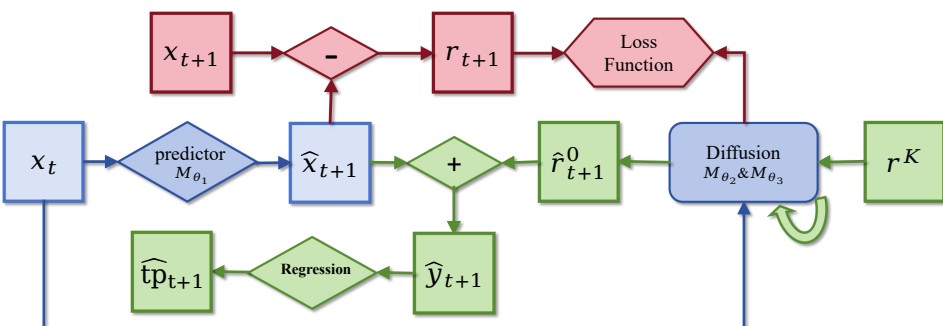

Figure 9: An illustration of the training and inference pipelines of ResCast. The blue part of Figure represents operations common to both the training and inference phases. The red part are used only during training, while the green part indicate processes exclusive to inference. The detailed processes are displayed in the extended view of Fig. 2

where $r^k = \sqrt{\alpha_k} r^0 + \sqrt{1 - \alpha_k} \epsilon$. After training, we can recover $r^{k-1}$ from $r^k$ with

$$r^{k-1} = \frac{1}{\sqrt{\alpha_k}} \left( r^k - \frac{\beta_k}{\sqrt{1 - \alpha_k}} \epsilon_\theta(r^k, k) \right) + \sigma_k \epsilon \tag{15}$$

where $\sigma_k$ is a variance hyperparameter. Finally, by drawing a sample from the prior distribution $p(r^K)$ and iteratively applying Eq. 15, a sample from the target distribution $p(r^0)$ can be derived.

## C.2 ILLUSTRATIVE SUMMARY AND ALGORITHM

In this section, we discuss the overall framework of ResCast, which extends the original diffusion-based theory of directly predicting precipitation residuals, providing a novel forecasting paradigm for medium-range precipitation prediction. Sec. 3 provides a detailed discussion of the technical aspects of ResCast. As shown in Fig. 9, the model integrates the training of diffusion and the entire precipitation inference process. Similar to the conditional diffusion model foundations in works (Yu et al., 2024) (Srivastava et al., 2024), ResCast employs a residual pipeline. It involves deterministic predictions corrected by the residuals generated from the conditional diffusion model.

**Training and Inference.** As shown in Fig. 2, ResCast is trained end-to-end, where both the direct weather predictor and the residual diffusion model are optimized within the same training iterations. The complete training process is summarized in Algorithm 1. During the inference phase, the framework first generates weather predictions with residuals, followed by precipitation regression. The inference process is summarized in Algorithm 2.

## C.3 CONSERVATION LAW LOSS

In the context of physically constrained neural network modeling, mass, energy, and momentum are conserved throughout the system's evolution. Traditional neural network models may violate these conservation laws during prediction, leading to physically inconsistent results. To address this issue, a conservation law constraint term can be introduced into the loss function as a regularization term, penalizing parts of the model's predictions that violate the conservation laws, thereby guiding the model to learn solutions that adhere to physical principles.

To justify the inclusion of the conservation loss $\mathcal{L}_{\text{conservation}}$ in Eq. 5, we provide a theoretical derivation that connects the residual-based modeling process with physical conservation principles in the context of atmospheric variables.

Let the predicted residual be defined as:

$$r_{t+1} = x_{t+1} - \mathcal{M}_{\theta_1}(x_t) \tag{16}$$

where $x_{t+1}$ is the observed future state of a meteorological variable (e.g., temperature, wind, humidity), and $\mathcal{M}_{\theta_1}(x_t)$ is the deterministic forecast given by the frozen weather predictor. The residual $r_{t+1}$ captures the discrepancy between observation and prediction.

**Algorithm 1** Training of The Residuals Diffusion Component

1: **Input:** Initial weather data $x_t$, noise $\epsilon \sim \mathcal{N}(0, I)$, model parameters $\theta_1, \theta_2, \theta_3$
2: Initialize model parameters $\theta_1, \theta_2, \theta_3$
3: **while** not converged **do**
4:      Getting direct prediction $\hat{x}_{t+1} = \mathcal{M}_{\theta_1}(x_t)$
5:      Build residual sequence $r_{t+1} = x_{t+1} - \hat{x}_{t+1}$
6:      Sample a sequence $(r, \hat{r}) \sim \mathbb{R}^{C \times H \times W}$, where $r$ residuals of the base climate variables
7:      Extracting global trend information $C_t$
8:      Sample diffusion step $k \sim \mathcal{R}(0, \ldots, K)$
9:      $\mathcal{L} = 0$
10:      Disturbing $r^0$ to $r^k$ following Eq. 11
11:      Compute the loss: $\mathcal{L}$ following Eq. 6
12:      Update model parameters: $(\theta_1, \theta_2, \theta_3) \leftarrow (\theta_1, \theta_2, \theta_3) - \nabla_{\theta_1, \theta_2, \theta_3} \mathcal{L}$
13: **end while**

**Algorithm 2** Inference of The Framework for Precipitation

1: **Input:** Initial weather data $x_t$
2: **Output:** Predicted precipitation $\hat{tp}_{t+1}$
3: Getting direct prediction $\hat{x}_{t+1} = \mathcal{M}_{\theta_1}(x_t)$
4: Extracting global trend information $C_t$
5: **while** Reverse diffusion from $k = K$ to $k = 1$ **do**
6:      Initial noise $\epsilon \sim \mathcal{N}(0, I)$
7:      Estimating target noise $\epsilon_{\theta_2}$ following Eq. **??**
8:      Recovering $r_{t+1}^{k-1}$ from $r_{t+1}^k$ following Eq. 15
9: **end while**
10: Calculate Final Weather data $\hat{y}_{t+1} = \hat{x}_{t+1} + \hat{r}_{t+1}$
11: Regression precipitation $\hat{tp}_{t+1}$ following Eq. 10

Now consider a general conservation law in fluid dynamics for a scalar quantity $x$, such as mass or energy:

$$\frac{\partial x}{\partial t} + \nabla \cdot \mathbf{F}(x) = S \tag{17}$$

where $\mathbf{F}(x)$ is the flux and $S$ is a source term. For data-driven modeling, we do not directly compute $\nabla \cdot \mathbf{F}(x)$, but we can enforce that the *change in state* is approximately conserved:

$$x_{t+1} \approx \mathcal{M}_{\theta_1}(x_t) + r_{t+1} \tag{18}$$

Substituting, the residual is:

$$r_{t+1} \approx x_{t+1} - \mathcal{M}_{\theta_1}(x_t) \tag{19}$$

We interpret $r_{t+1}$ as an approximation of the physical deviation governed by unmodeled dynamics. To ensure it satisfies conservation over a spatiotemporal domain, we require the *sum of residuals across samples* to be approximately conserved. Hence, we define the following conservation constraint:

$$\mathcal{L}_{\text{conservation}} = \frac{1}{N} \sum_{i=1}^{N} \left( x_{t+1} - \mathcal{M}_{\theta_1}(x_t) \right)_i^2 = \frac{1}{N} \sum_{i=1}^{N} (r_{t+1})_i^2 \tag{20}$$

Minimizing this term ensures that the total predicted residuals stay close to zero-mean and physically bounded, consistent with the underlying conservation dynamics. This loss thus acts as a *soft regularization* that steers the diffusion-generated residuals to obey first-order conservation behavior.

# D DATASETS AND DATA PREPARATION

## D.1 ERA5

The experimental dataset used in this work is the **ERA5** dataset (Hersbach et al., 2018) from the European Centre for Medium-Range Weather Forecasts, available at https://dataserv.ub.tum.de/index.php/s/m1524895. This dataset contains ERA5 data from 1979 to 2018, including 6 upper-level variables at 13 levels, 8 surface variables, and 4 constant variables. The dataset has three resolution options: 5.625° ($32 \times 64$), 2.8125° ($64 \times 128$), and 1.40625° ($128 \times 256$). For improved accuracy in medium-range precipitation predictions, we selected the data with a resolution of 1.40625°. Furthermore, 22 variables were selected for precipitation regression, as shown in Table 10. To mitigate the issue of error accumulation in medium-range predictions, the data underwent appropriate preprocessing.

In the preprocessing stage, we processed the hourly ERA5 data (both surface and upper-level variables) into daily averages with a 6-hour time interval. Specifically, we first aggregated the data by summing and averaging the values over a 24-hour period, effectively reducing the original 24 hourly data points to 4 values per day (with a 6-hour time gap). However, certain variables require special handling, including wind speed and precipitation.

For wind speed, the original data represents both magnitude and direction. To obtain a meaningful daily value, we first computed the vector magnitude of the wind in both the $u$- and $v$-dimensions (representing the east-west and north-south components of wind, respectively) for each hour. This is done by summing the squares of the $u$ and $v$ components, taking the square root, and then performing the $T$-hour ($T = 24$) aggregation:

$$\text{Wind}_t = \frac{1}{T} \sum_{t_0=t}^{t+T} \sqrt{u_{t_0}^2 + v_{t_0}^2} \tag{21}$$

For precipitation, the data was accumulated over the 24-hour period without averaging, as the total daily precipitation is of primary interest. The precipitation for each day can be expressed as:

$$tp_t = \sum_{t_0=t}^{t+T} P_{t_0} \tag{22}$$

This approach ensures that both wind speed and precipitation are appropriately processed, reflecting their specific physical characteristics. After this preprocessing step, the data is ready for use in the subsequent stages of the model. Additionally, the dataset was divided into training (1979-2015), validation (2016), and testing (2017-2018) sets based on years.

### D.2 DATASET NORMALIZATION

In this work, dataset preprocessing includes a normalization step that ensures each variable is appropriately scaled for input into the model. This normalization process is crucial for optimizing model performance, especially for variables with different units or magnitudes. The preprocessing involves calculating the mean and standard deviation (std) for each variable and applying normalization based on these statistics.

For each variable, the mean and standard deviation are computed across the training set. Denote the variable of interest by $x$, its mean by $\mu_x$, and its standard deviation by $\sigma_x$. The normalization is performed using the following formula:

$$x_{norm} = \frac{x - \mu_x}{\sigma_x} \tag{23}$$

where $x_{norm}$ represents the normalized variable. This step ensures that all input features have zero mean and unit variance, which accelerates convergence during model training.

In the case of precipitation data, a specific transformation is applied to stabilize the variance and reduce the effect of extreme values. Following the approach outlined in (Pathak et al., 2022), we apply a log transformation to the precipitation variable $tp$, defined as:

$$tp_{norm} = \log(1 + \frac{tp}{\epsilon}) \tag{24}$$

where $\epsilon = 1 \times 10^{-5}$. Since total precipitation values are highly sparse, this transformation discourages the network from predicting zeros and ensures a less skewed distribution of values. After normalization, the processed data is ready to be fed into the model, with each variable appropriately scaled and transformed to facilitate learning. The combination of normalization and transformation ensures that the model can effectively capture patterns in the data without being dominated by any particular variable's scale.

### D.3 WEATHERBENCH2

WeatherBench2 (Rasp et al., 2024) is a framework for evaluating and comparing data-driven and traditional numerical weather forecasting models. The WeatherBench2 dataset is available at

https://console.cloud.google.com/storage/browser/weatherbench2. From this work, the baselines can be described as follows:

- **ENS mean** is a numerical weather model that provides ensemble forecasts with 50 members, offering multiple outcomes for medium-range predictions (up to 15 days). In the experiments, the ensemble mean of these member forecasts is used as a deterministic forecast for comparison.
- **HERS** is a system developed by ECMWF that provides high-resolution deterministic forecasts for improving medium-range weather predictions (up to 10 days).
- **GraphCast** (Lam et al., 2022) is a machine learning-based model that constructs graph networks, outperforming traditional models (up to 10 days).

The predicted values for the above models are downloaded at a resolution of 0.25°. To ensure a fair comparison, the data was resampled to a uniform resolution of 1.40625° for the evaluation of metrics.

### D.4 MODEL REPRODUCTION AND INFERENCE

**FourCastNet** (Pathak et al., 2022) employs a fully data-driven machine learning framework that leverages Fourier integral operators to model the spatiotemporal evolution of multiple atmospheric variables. After forecasting these fields, it applies a lightweight regression head to produce precipitation estimates out to 10 days. The source code is publicly available at https://github.com/NVlabs/FourCastNet. In our work, we have successfully reproduced the 0.25° resolution 10-day precipitation inference for the year 2018.

Additionally, the **GenCast** (Price et al., 2025) implementation is open-sourced alongside GraphCast at https://github.com/google-deepmind/graphcast. We reproduced a representative small-scale inference case from GenCast and conducted a direct comparison with our model.

## E MODELS SETTING AND VARIABLE SELECTION EXPERIMENT

For the precipitation regression task, we use the following models, including our Precipitation Regression Model, within the same training framework. In the following subsection, we will elaborate on the details of the model architecture implementation and the experimental results.

### E.1 HYPERPARAMETERS OF EACH TRAINING FRAMEWORK

#### E.1.1 PROPOSED FRAMEWORK

The model was trained using the hyperparameters listed in Table 4. This configuration led to significant improvements in performance during the training process, contributing to better results in precipitation prediction tasks.

Table 4: Training Hyperparameters of our Framework

| Hyperparameter | Description | Typical Value |
|---|---|---|
| Optimizer | Type of optimizer used | AdamW (Kingma, 2014) |
| $\lambda_1$ | First momentum parameter | 0.9 |
| $\lambda_2$ | Second momentum parameter | 0.99 |
| Weight Decay | Weight decay applied to all parameters | 1e-5 |
| Learning Rate | Initial learning rate | 5e-5 |
| Linear Warmup Step | the steps of Linear Warmup | 5000 |
| Cosine Annealing Step | the steps of Cosine Annealing | 95000 |

#### E.1.2 U-NET

We directly use the open-source U-Net (Ronneberger et al., 2015) architecture, with the only modification being the regression channel number from $C$ to 1. Other hyperparameters are as shown

in Table 5. The framework is somewhat analogous to semantic segmentation from a color map to a single layer.

Table 5: The hyperparameters of U-Net (for Semantic Segmentation)

| Hyperparameter | Description | Typical Value |
|---|---|---|
| Input Image Size | Dimensions of input images | $128 \times 256$ |
| Kernel Size | Size of convolutional kernels | 3 |
| Number of Convolutional Layers | Layers per encoder/decoder block | 2 |
| Number of Channels | Output channels per layer | (64,128,256) |
| Pooling Size | Max pooling layer size | 2 |
| Activation Function | Activation function used | ReLU |
| Output Channels | Number of output channels (classes) | 1 |
| Learning Rate | Initial learning rate | 1e-3 |
| Batch Size | Number of samples per batch | 32 |

### E.1.3 VIT

We adopt the publicly available Vision Transformer (ViT) (Dosovitskiy et al., 2020) as our backbone, modifying only the prediction head for regression by reducing its output dimensionality from $C$ channels to a single channel. All other hyper-parameters remain unchanged and are summarized in Table 6.

Table 6: The hyperparameters of Vision Transformer (ViT)

| Hyperparameter | Description | Typical Value |
|---|---|---|
| Input Image Size | Dimensions of input images | $128 \times 256$ |
| Patch Size | Size of each image patch | 4 |
| Embedding Dimension | Dimension of patch embeddings | 256 |
| Number of Layers | Number of transformer encoder layers | 12 |
| Number of Heads | Number of attention heads per layer | 12 |
| MLP Ratio | Expansion ratio in MLP layers | 4 |
| Dropout Rate | Dropout rate for regularization | 0.1 |
| Learning Rate | Initial learning rate | 1e-4 |
| Batch Size | Number of samples per batch | 32 |

### E.1.4 AFNO

We adopted the precipitation regression module from AFNO in FourCastNet (Pathak et al., 2022), which is currently a popular and effective network component for regressing precipitation results. We conducted experiments using the hyperparameters set as shown in Table 7.

Table 7: The hyperparameters for AFNO Precipitation Regression Module

| Hyperparameter | Description | Typical Value |
|---|---|---|
| Batch Size | Size of global training batch | 16 |
| Learning Rate | Initial learning rates | 2.5e-4 |
| Patch Size | Size of each image patch | 4 |
| Sparsity Threshold $\lambda$ | Regularization threshold for sparsity | 1e-2 |
| Number of AFNO Blocks $n_b$ | Number of blocks in the AFNO module | 8 |
| Depth | Number of layers in the transformer | 12 |
| MLP Ratio | Expansion ratio in MLP layers | 4 |
| Embedding Dimension | Dimension of the embedding in AFNO | 256 |
| Dropout | Dropout rate for regularization | 0 |

### E.1.5 PRECIPITATION REGRESSION MODEL

The Precipitation Regression method is the approach proposed in this work. As shown in Table 8, we employ the following default hyperparameters for Precipitation Regression Module in all of our experiments.

Table 8: The hyperparameters of Precipitation Regression Model

| Hyperparameter | Description | Value |
|---|---|---|
| Input Image Size | Dimensions of input images | $128 \times 256$ |
| Kernel Size | Size of convolutional kernel | 3 |
| Patch Size | Size of each input patch | 4 |
| Embedding Dimension | Patch embedding dimension | 256 |
| MLP Ratio | MLP hidden dimension ratio | 4 |
| Dropout Rate | Dropout rate for regularization | 0.1 |
| Learning Rate | Initial learning rate | 1e-4 |

Table 9: Experimental Results for Variable Selection in Precipitation Regression

| Input Variables | Input Levels | RMSE↓ | MAE↓ | ACC↑ | F1@0.1mm↑ | Training Time |
|---|---|---|---|---|---|---|
| Wind, T, Q, R | 50, 100, 150, 200, 300, 400, 500, 700, 850, 1000 | 6.5602 | 2.3906 | 0.7155 | 0.7422 | ˜12h30m |
| Z, Wind, Q, R | 50, 100, 150, 200, 300, 400, 500, 700, 850, 1000 | 6.5334 | 2.5223 | 0.7133 | 0.7300 | ˜12h20m |
| Z, Wind, T, Q, R | 50, 100, 150, 200, 300, 400, 500, 700, 850, 1000 | **6.0861** | **2.2292** | **0.7646** | 0.7562 | ˜14h40m |
| Z, Wind, T, Q, R | 200, 300, 400, 500, 700, 850, 1000 | 6.0998 | 2.2479 | 0.7532 | 0.7489 | ˜10h10m |
| Z, Wind, T, Q, R | 500, 700, 850, 1000 | 6.0879 | 2.2511 | 0.7624 | **0.7589** | ˜9h30m |

## E.2 SELECTION EXPERIMENT

In precipitation forecasting research, atmospheric data from different pressure levels can capture key circulation features across multiple spatiotemporal scales (Dong et al., 2024). This paper proposes an innovative residual-model forecasting framework, using a Diffusion-based Spatiotemporal network to predict more accurate weather fields for precipitation regression. Specifically, meteorological variables data from the mid-to-upper levels, such as 500 hPa and 700 hPa, can capture large-scale circulation patterns, providing key input features for the regression task in precipitation forecasting. These layers are significant indicators of cyclonic activity and moisture transport (Cavazos & Hewitson, 2002; Liu et al., 2019). On the other hand, low-level atmospheric data, such as from 850 hPa and 1000 hPa, is closely related to surface weather systems and fronts, providing local precipitation information needed for the regression model (Holton & Hakim, 2013).

We used our proposed Precipitation Regression Module to conduct the variable selection experiments, primarily focusing on identifying which variables and corresponding levels contribute to precipitation regression. The experiments were trained on eight GPUs with 80GB of memory, achieving 77.6 TFLOPS of computing power. The model was trained for 30 epochs. The parameter settings of the model are shown in Table 8.

### E.2.1 COMPARISON RESULTS

Since surface variables are relatively few and have a minor impact on computational complexity, we primarily focus on the effect of upper-level variables. Previous studies have demonstrated that variables such as Wind, Q, and R are significantly beneficial and highly correlated with precipitation distribution (Cheng et al., 2025). Therefore, ablation experiments were not conducted for these three variables. The experiment involved training with different upper-level variables and variable levels as inputs. To evaluate the effect of selecting experimental variables, we formed five control groups and calculated various precipitation metrics as references. The results are shown in Table 9.

Table 10: Dataset Variable List for Different Types and Levels.

| Type | Variable name | Abbrev. | Levels |
|------|---------------|---------|--------|
| Static | Land-sea mask | LSM | - |
| Static | Orography | - | - |
| Surface | 2 metre temperature | T2m | - |
| Surface | 10m Wind speed | $Wind_{10}$ | - |
| Upper | Geopotential | Z | 500, 700, 850, 1000 |
| Upper | Wind speed | Wind | 500, 700, 850, 1000 |
| Upper | Temperature | T | 500, 700, 850, 1000 |
| Upper | Specific humidity | Q | 500, 700, 850, 1000 |
| Upper | Relative humidity | R | 500, 700, 850, 1000 |

### E.2.2 ANALYSIS AND SUMMARY

From the Table 9, it is clear that reducing number of upper-level variables has a significant impact on the precipitation metrics. The absence of these variables leads to a noticeable decrease in key performance indicators such as RMSE, MAE, and F1 score. This demonstrates that upper-level variables are crucial for accurate precipitation prediction. However, varying the number of levels (e.g., atmospheric pressure levels) has a relatively minor effect on the results. This suggests that while the inclusion of upper-level variables is important for improving accuracy, reducing the number of levels does not substantially affect the regression task. As a result, using fewer levels helps decrease model inference time and computational requirements without sacrificing prediction quality.

Therefore, the dataset variables we finally selected are shown in Table 10.

### E.3 METRICS

In this section, we detail the evaluation metrics used in our experiments. LPIPS and SSIM are used to assess the visual quality of meteorological variable residuals $\hat{r} \in \mathbb{R}^{C \times H \times W}$. The remaining metrics are applied to evaluate the precipitation predictions. The precipitation predictions and ground truth are formatted as $1 \times H \times W$.

**Learned Perceptual Image Patch Similarity (LPIPS)**   LPIPS is a perceptual similarity metric that measures the perceptual distance between two images, focusing on how human vision perceives differences in texture, color, and structure. In this work, we apply LPIPS to assess the visual quality of the residuals generated by the Residual Diffusion Component. The lower values indicating higher perceptual similarity between the predicted residuals to the ground truth residuals.

$$\text{LPIPS}(\hat{r}, r) = \sum_l \frac{1}{H_l W_l} \| w_l \odot (\phi_l(\hat{r}) - \phi_l(r)) \|_2^2 \tag{25}$$

where $\phi_l(\hat{r})$ and $\phi_l(r)$ represent the features at the $l$-th layer for the predicted and true residuals. $w_l$ is the learned weight for the $l$-th layer. $\odot$ denotes element-wise multiplication, and $\| \cdot \|_2$ is the L2 norm used to measure the difference.

**Structural Similarity Index Measure (SSIM)**   SSIM is a well-known metric for measuring the similarity between two images, focusing on structural information such as luminance, contrast, and texture. It is commonly used to assess image quality in various applications, and in this work, SSIM is applied to evaluate the structural quality of the generated weather variable residuals. A higher SSIM score indicates that the predicted residuals closely resemble the ground truth in terms of structure and spatial relationships.

$$\text{SSIM}(\hat{r}, r) = \frac{(2\mu_{\hat{r}}\mu_r + c_1)(2\sigma_{\hat{r}r} + c_2)}{(\mu_{\hat{r}}^2 + \mu_r^2 + c_1)(\sigma_{\hat{r}}^2 + \sigma_r^2 + c_2)} \tag{26}$$

where $\mu_{\hat{r}}$ and $\mu_r$ are the mean values of the predicted and true residuals. $\sigma_{\hat{r}}^2$ and $\sigma_r^2$ are the variances of the predicted and true residual. $\sigma_{\hat{r}r}$ is the covariance between the predicted and true residuals, and $c_1$ and $c_2$ are constants used to stabilize the denominator and avoid division by zero.

**Root mean square error (RMSE)**  Root Mean Square Error (RMSE) evaluates the average size of the errors in a dataset. It measures how well the predicted values align with the true values, where a smaller RMSE indicates better prediction accuracy.

$$\text{RMSE} = \frac{1}{N} \sum_{k=1}^{N} \sqrt{\frac{1}{H \times W} \sum_{i=1}^{H} \sum_{j=1}^{W} L(i)(\hat{tp}_{k,i,j} - tp_{k,i,j})^2} \tag{27}$$

where $L(i)$ is introduced as a latitude-based weighting factor:

$$L(i) = \frac{\cos(\text{lat}(i))}{\frac{1}{H} \sum_{i'=1}^{H} \cos(\text{lat}(i'))} \tag{28}$$

**The spatial distribution of RMSE**  The spatial distribution of Root Mean Square Error (RMSE) provides a comprehensive view of model performance across different geographical regions. For each grid point (i,j), the error is weighted by the cosine of latitude to account for the decreasing grid cell area towards the poles, ensuring a fair comparison across different latitudes.

**Anomaly correlation coefficient (ACC)**  The Anomaly Correlation Coefficient (ACC) evaluates the accuracy of a forecast by quantifying the correlation between predicted and observed anomalies. It is widely used in climate and weather prediction to assess how effectively the model identifies deviations from the climatological mean. A higher ACC value signifies better forecasting performance.

$$\text{ACC} = \frac{\sum_{k,i,j} L(i)\hat{TP}_{k,i,j} TP_{k,i,j}}{\sqrt{\sum_{k,i,j} L(i)\hat{TP}_{k,i,j}^2 \sum_{k,i,j} L(i)TP_{k,i,j}^2}} \tag{29}$$

$$\hat{TP}_{k,i,j} = \hat{tp}_{k,i,j} - C_{k,i,j}, \quad TP_{k,i,j} = tp_{k,i,j} - C_{k,i,j} \tag{30}$$

where Climatology C represents the average of the ground truth data across the entire test set over time. The calculation of $L(i)$ is equivalent to the formula given in Eq. 28.

**F1 Score**  The F1 score is a commonly used metric for evaluating the performance of binary classification models, particularly in imbalanced datasets. It is the harmonic mean of precision and recall, providing a balance between the two. A higher F1 score indicates better performance, with values ranging from 0 (worst) to 1 (best). The F1 score is especially useful when the positive and negative classes are imbalanced, as it accounts for both false positives and false negatives.

$$F1 = \frac{2 \cdot \text{Precision} \cdot \text{Recall}}{\text{Precision} + \text{Recall}} \tag{31}$$

where precision and recall are defined as:

$$\text{Precision} = \frac{TP}{TP + FP}, \quad \text{Recall} = \frac{TP}{TP + FN} \tag{32}$$

$$TP = \sum_{k,i,j} \left( (\hat{tp}_{k,i,j} \geq th) \wedge (tp_{k,i,j} \geq th) \right)$$
$$FP = \sum_{k,i,j} \left( (\hat{tp}_{k,i,j} \geq th) \wedge (tp_{k,i,j} < th) \right) \tag{33}$$
$$FN = \sum_{k,i,j} \left( (\hat{tp}_{k,i,j} < th) \wedge (tp_{k,i,j} \geq th) \right)$$

Here, $TP$ represents the number of true positives, $FP$ is the number of false positives, and $FN$ is the number of false negatives. In this paper, $th$ represents the precipitation threshold, which is typically set to 0.1mm to determine whether rain is occurring.

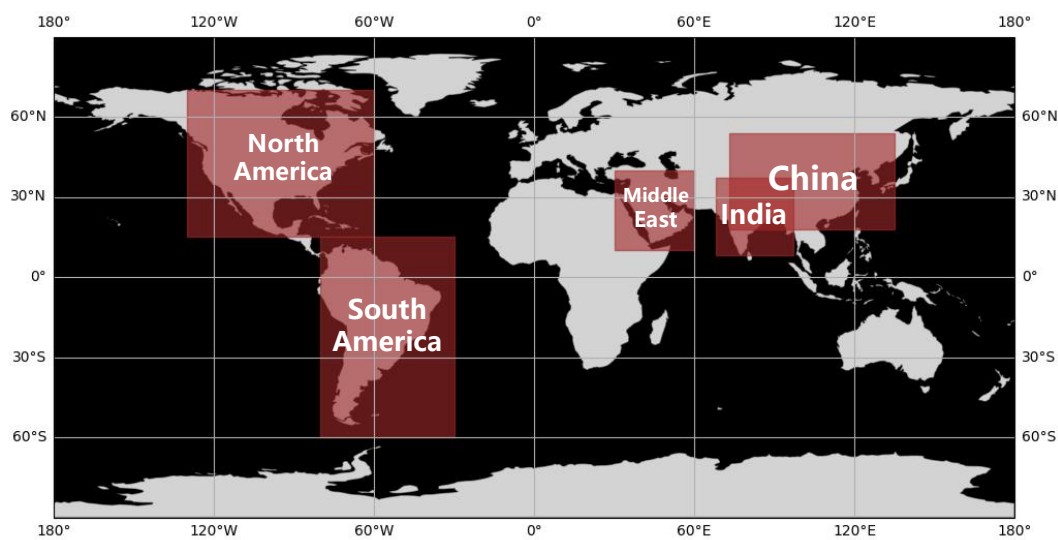

Figure 10: **Region specification for analysis.** Per-region evaluation is provided in Fig. 4.

**Threat Score (TS)**  The Threat Score (TS), also known as the Critical Success Index (CSI), is a metric commonly used in meteorology and weather forecasting. It assesses the accuracy of a forecast by measuring the overlap between observed and predicted positive events. It is particularly useful in evaluating the prediction of rare or extreme events. The TS score ranges from 0 (worst) to 1 (best), with higher values indicating better performance. The TS score penalizes both false positives and false negatives, providing a more balanced evaluation of the model's performance when predicting rare events.

$$TS = \frac{TP}{TP + FP + FN} \tag{34}$$

where $TP$, $FP$, and $FN$ are defined in the same way as in Eq. 33. Here, $TP$ represents the number of true positives, $FP$ is the number of false positives, and $FN$ is the number of false negatives. In this paper, $th$ represents the precipitation threshold, which is typically set to 0.1mm to determine whether rain is occurring.

**Energy Conservation Metric (ECM)**  The Energy Conservation Metric (ECM) compares the energy ratio between the generated data with diffusion residuals (forecast) and the real ERA5 data. Additionally, it also compares **the energy ratio** between the direct prediction (direct data) and the real ERA5 data. The ECM evaluates the degree to which the generated or direct prediction data conserves energy when compared to the actual observed data, offering a useful performance metric in weather forecasting and climate models.

The ECM is computed using the following formula:

$$TE = KE + PE + IE + LH = \frac{1}{2}(u^2 + v^2) + gz + c_pT + L_vq \tag{35}$$

where $TE$ represents the total energy, composed of:

- **Kinetic Energy (KE)**: $\frac{1}{2}(u^2 + v^2)$,
- **Potential Energy (PE)**: $gz$,
- **Internal Energy (IE)**: $c_pT$,
- **Latent Heat (LH)**: $L_vq$.

where, $u$ and $v$ represent the wind components, $z$ is the geopotential height, $T$ is the temperature, and $q$ is the specific humidity. $c_p$ is the specific heat at constant pressure, and $L_v$ is the latent heat of vaporization. The physical constants used in the computation are as follows:

- Gravitational acceleration $g = 9.81\,\text{m/s}^2$,
- Specific heat $c_p = 1004.6\,\text{J/(kg·K)}$,
- Latent heat of vaporization $L_v = 2.5 \times 10^6\,\text{J/kg}$.

By comparing the ECM of both the generated and direct prediction data with the real ERA5 data, the closer the ECM value is to 1, the more accurate the generated or direct prediction data is in terms of total energy conservation.

## F  DISCUSSION

Despite its strong performance, ResCast has several limitations. First, the model's spatial resolution is limited to $1.40625°$, which can smooth out localized precipitation extremes, especially in complex terrain. Second, our framework relies solely on ERA5 reanalysis data, potentially overlooking real-time observational uncertainties and biases. Third, the deterministic design of ResCast precludes uncertainty quantification; extending to probabilistic or ensemble-based forecasts remains future work. Fourth, training costs constrained optimization: we stopped training at 10,000 steps due to computational budget, although the loss curve indicates further gains are possible with additional training and hyperparameter tuning. Finally, our current variable set may omit key meteorological factors directly impacting precipitation; identifying and incorporating these additional predictors could further improve forecast accuracy.

**Enhanced Forecast Accuracy for Disaster Risk Reduction**  ResCast offers several positive impacts, one of which is its ability to enhance forecast accuracy for disaster risk reduction. By improving 5–15 day precipitation forecasts by 3%–10% compared to existing methods, ResCast provides more reliable early warnings for extreme weather events such as floods, droughts, and other disasters. This improvement allows governments and emergency management agencies to allocate resources more effectively and develop more efficient defense strategies, reducing the risks associated with these events.

**Support for Agriculture and Water Resource Management**  Another positive impact is the support ResCast offers for agriculture and water resource management. The more accurate medium-range precipitation forecasts help farmers optimize irrigation schedules and planting decisions, thereby minimizing the risks of crop damage. Additionally, the model informs reservoir operations and watershed management, improving water use efficiency and promoting more sustainable management of water resources.

**Advancement of Machine Learning in Meteorology**  Moreover, the novel approach combining residual diffusion with Precipitation regression significantly advances machine learning in meteorology. This new paradigm can be extended to other meteorological variables and regional forecasts, opening up fresh avenues for AI applications in meteorology and environmental monitoring.

**Risk of Overreliance on Data-Driven Models**  However, there are several negative impacts and challenges associated with the model. One primary concern is the risk of overreliance on data-driven models. Since the model's performance heavily depends on reanalysis datasets like ERA5, any deficiencies in the data—especially with rare extreme events—can result in mispredictions. This limitation could potentially misguide decision-making in critical scenarios, where traditional forecasting models and expert analysis are crucial.

**Computational Resources and Environmental Impact**  Another negative aspect is the significant computational resources required to train and deploy ResCast. Our experiments, for example, used eight 80 GB GPUs, leading to high energy consumption and operational costs. This increased energy demand contributes to a higher carbon footprint, raising concerns about the environmental impact and sustainability of such large-scale models in operational settings.

**Lack of Interpretability and Transparency**    Furthermore, the lack of interpretability and transparency in the model poses a challenge. The complex structure of the diffusion model and multi-layer regression approach often results in a "black-box" system. This limited transparency may undermine meteorologists' trust in the model, particularly in high-stakes decision-making scenarios. To address this, it is essential to use ResCast in conjunction with traditional physical models and expert oversight, ensuring more reliable and interpretable outcomes.

# G    ADDITIONAL RESULTS

## G.1    ADDITIONAL WEATHER FORECASTING CASES AND METRIC RESULTS

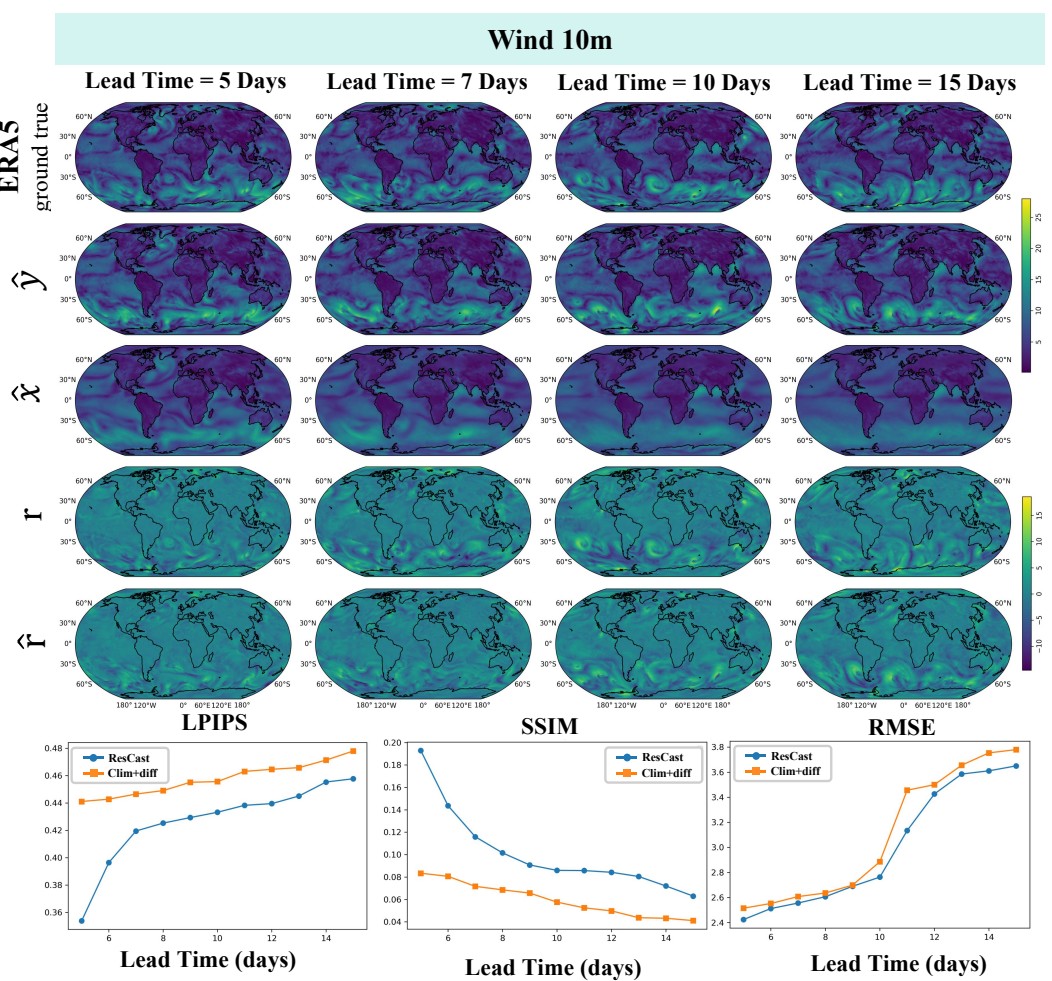

Figure 11: Visual examples of predictions $\hat{y}, \hat{x}$ and residuals $r, \hat{r}$, along with performance metrics (LPIPS↓, SSIM↑, RMSE↓) plotted over varying lead times for Wind 10m. Forecast example initialized at 2018-05-31 00:00 UTC, with plots corresponding to 5, 7, 10 and 15 day lead times.

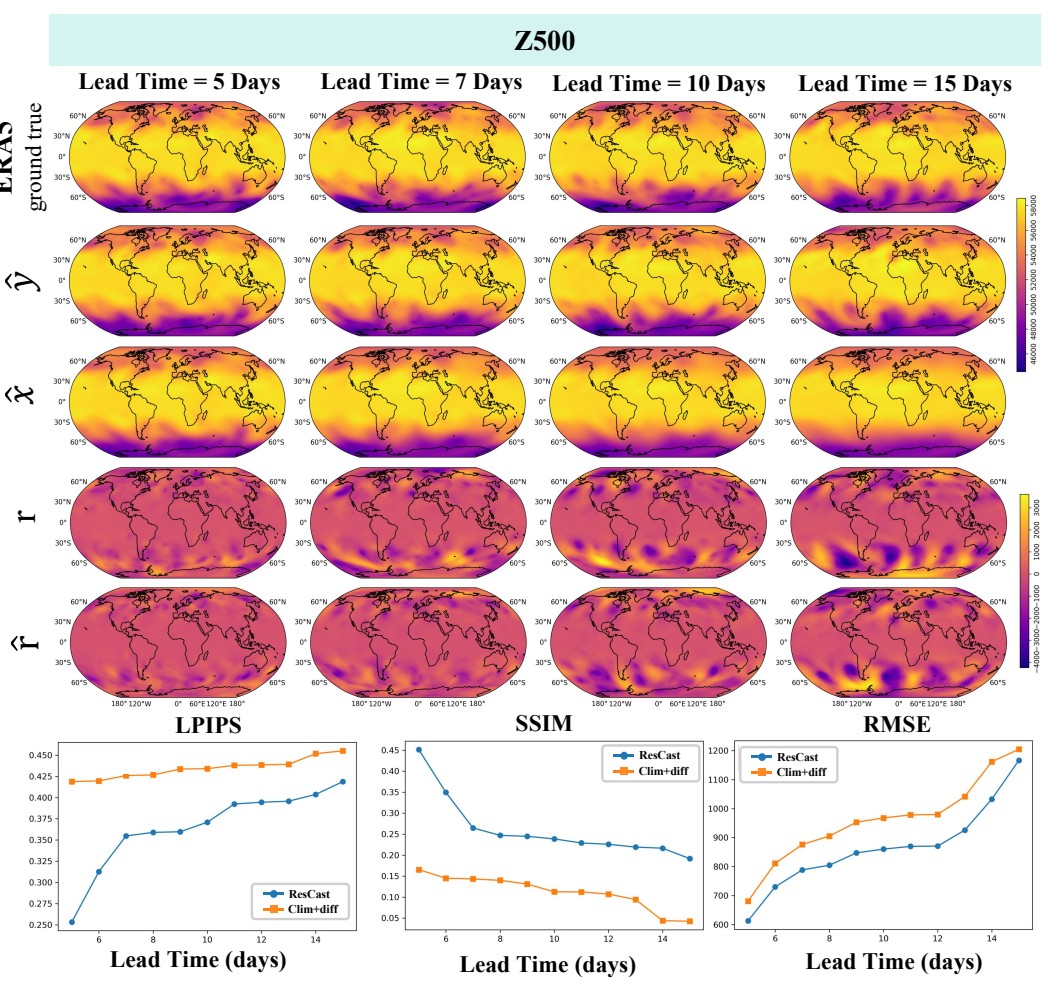

Figure 12: Visual examples of predictions $\hat{y}, \hat{x}$ and residuals $r, \hat{r}$, along with performance metrics (LPIPS↓, SSIM↑, RMSE↓) plotted over varying lead times for Z500. Forecast example initialized at 2018-05-31 00:00 UTC, with plots corresponding to 5, 7, 10 and 15 day lead times.

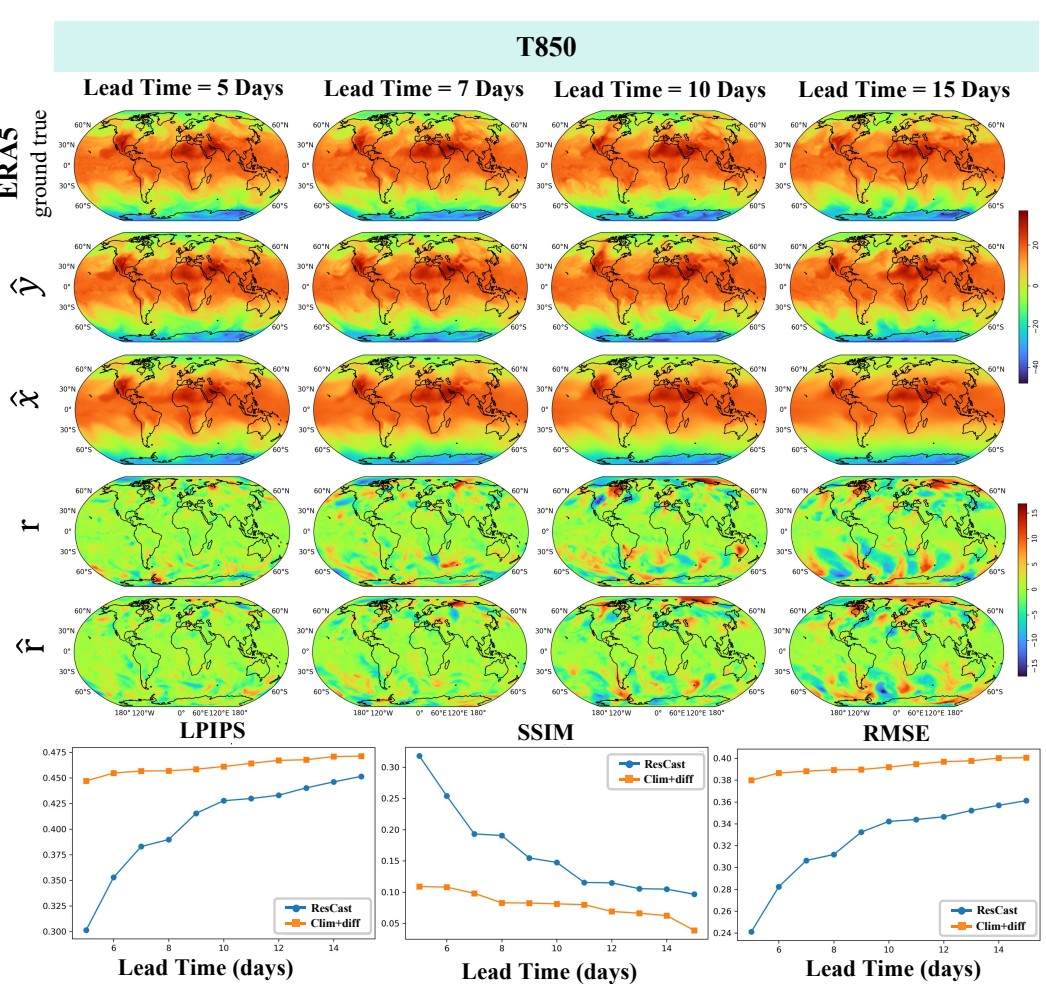

Figure 13: Visual examples of predictions $\hat{y}, \hat{x}$ and residuals $r, \hat{r}$, along with performance metrics (LPIPS↓, SSIM↑, RMSE↓) plotted over varying lead times for T850. Forecast example initialized at 2018-05-31 00:00 UTC, with plots corresponding to 5, 7, 10 and 15 day lead times.

## G.2   PERFORMANCE COMPARISON AND CASE STUDY OF DIFFERENT PRECIPITATION FORECASTING MODELS

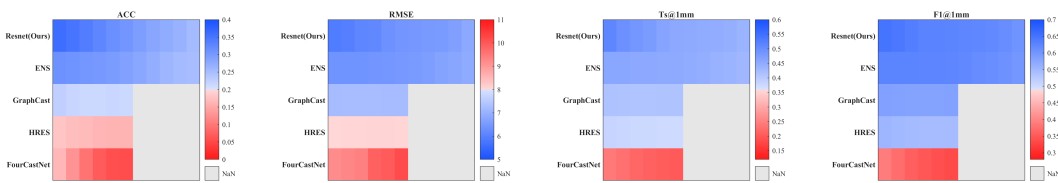

Figure 14: Performance comparison of different precipitation forecasting models across multiple metrics and lead times. The heatmaps show four key metrics (ACC, RMSE, Ts@1mm, and F1@1mm) evaluated from 5 to 15 hours lead time. Our ResNet-based model (top row) consistently outperforms other state-of-the-art models, including the operational ensemble forecast (ENS), GraphCast, HRES, and FourCastNet. Notably, our model achieves superior performance in all metrics: higher ACC (up to 0.36), lower RMSE (maintaining below 7.0), better threat score (Ts@1mm ¿ 0.45), and higher F1 score (exceeding 0.62) across all lead times. The performance advantage is particularly pronounced in longer lead times (¿10 hours), demonstrating robust and stable prediction capability. Gray areas indicate unavailable data for some models at extended lead times.

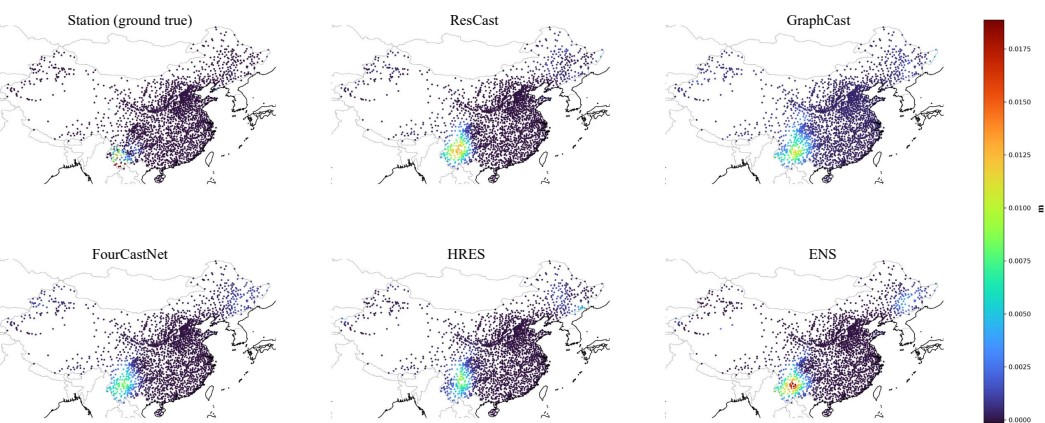

Figure 15: In these six panels, we compare station level precipitation forecasts obtained by interpolating grid outputs against ground truth. ResCast captures the spatial extent and intensity of heavy precipitation clusters more accurately than baselines, showing sharper and more localized peaks around the high-rainfall region. GraphCast and HRES reproduce general patterns but tend to underpredict peak intensities, while FourCastNet appears overly smooth and diffuses rainfall hotspots. ENS resolves local extremes better than FourCastNet but still misses finer structures that ResCast recovers. Overall, ResCast delivers the closest match to station observations, demonstrating superior detail and localization in point-based forecasts.

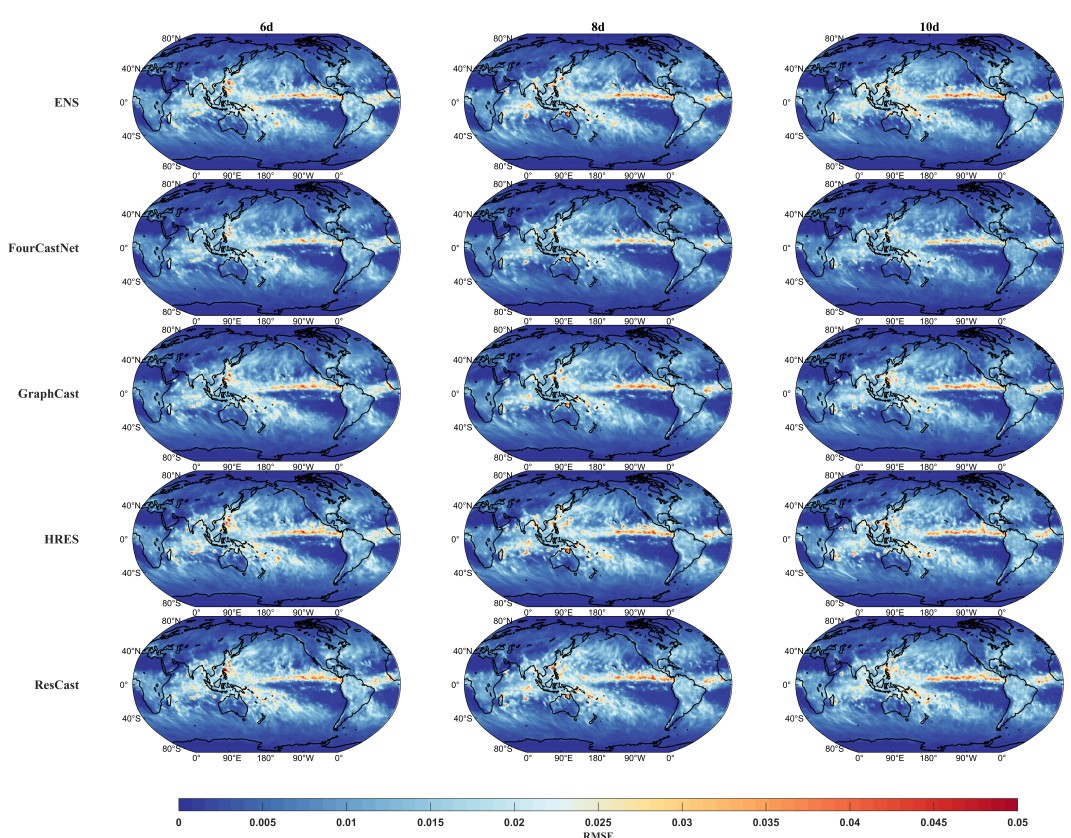

Figure 16: Spatial distribution of Root Mean Square Error (RMSE) for different forecast lead times (6, 8, and 10 days) across multiple models. The panels show RMSE values for ENS (ensemble mean of 50 ECMWF forecast members, first row), FourCastNet (second row), GraphCast (third row), HRES (ECMWF high-resolution forecast, fourth row), and ResCast (our model, bottom row). Each column represents a different forecast lead time (6, 8, and 10 days). The RMSE values range from 0 to 0.03, with lower values (darker blue) indicating better forecast accuracy. Higher RMSE values (yellow to red) are particularly visible in regions with high precipitation variability, such as the Intertropical Convergence Zone (ITCZ).

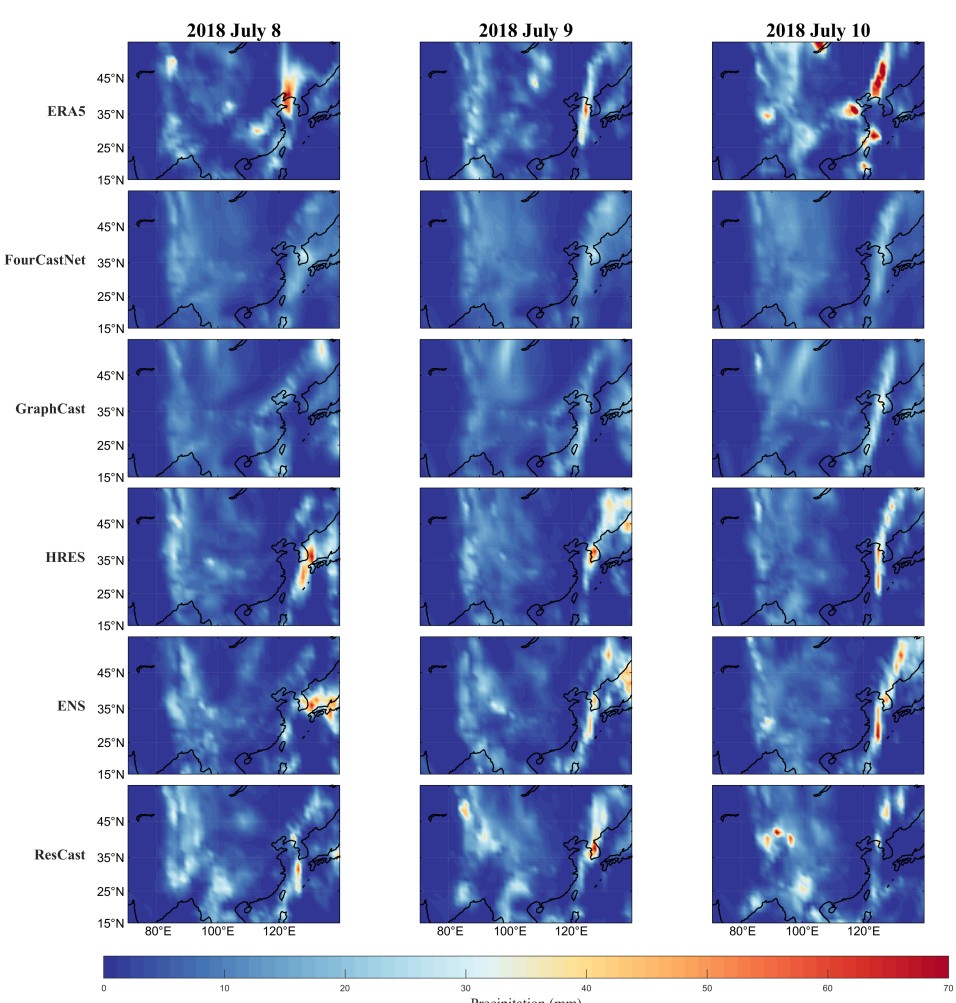

Figure 17: Comparison of precipitation forecasts for an extreme precipitation event over China during July 8-10, 2018. The panels show daily precipitation (mm) from different models: ERA5 (reanalysis data, first row), FourCastNet (second row), GraphCast (third row), HRES (ECMWF high-resolution forecast, fourth row), ENS (ensemble mean of 50 ECMWF forecast members, fifth row), and ResCast (our model, bottom row). Each column represents one day from July 8 to July 10, 2018. ResCast exhibits better skill in capturing both the spatial distribution and intensity of extreme precipitation events, particularly evident in the coastal regions of China. Compared to other models, ResCast more accurately represents the localized precipitation patterns and their temporal evolution, showing closer agreement with ERA5 reanalysis data.

### G.3 COMPARISON WITH GENCAST ON A SMALL-SCALE INFERENCE CASE

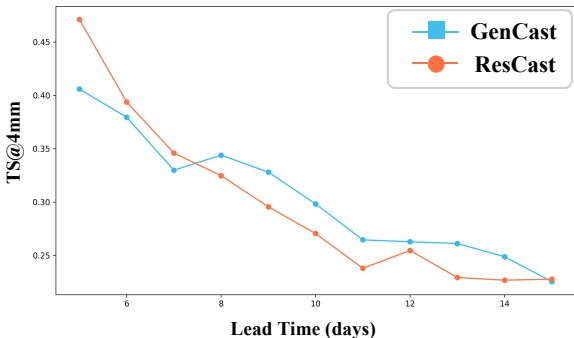

Figure 18: We evaluated the ts score at a 4 mm threshold for both our method and Gencast on the sample starting from March 29, 2019, across forecast lead times ranging from 5 to 15 days. Our method outperforms Gencast at lead times of 5 to 7 days, demonstrating superior short-term precipitation forecasting capability, but shows slightly inferior performance compared to Gencast at longer lead times.

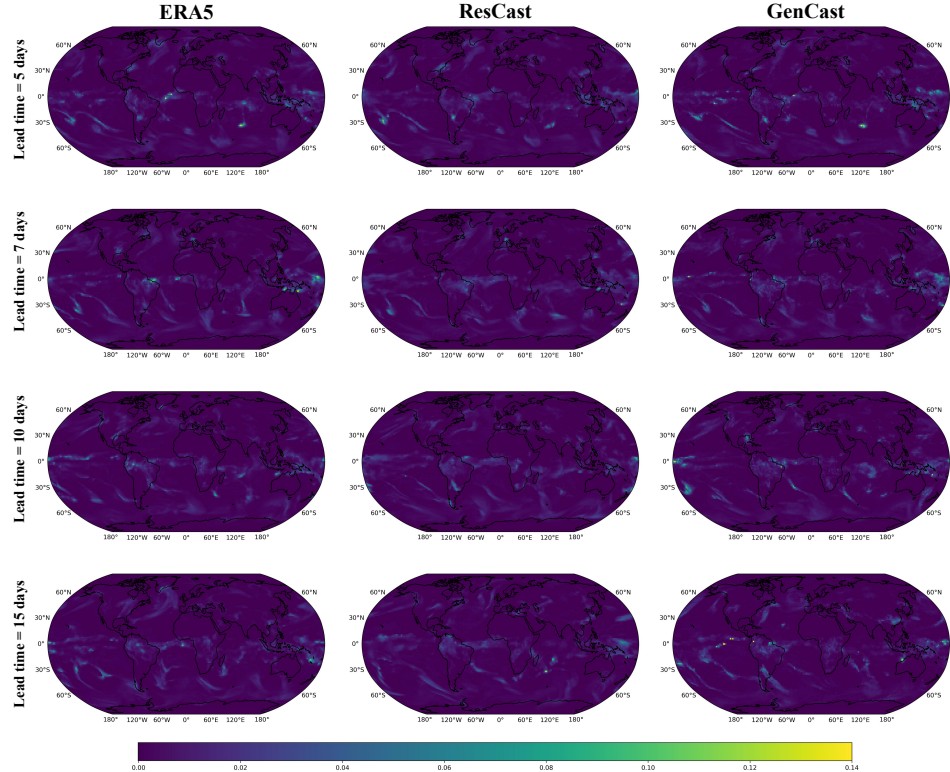

Figure 19: We reproduced the samples publicly provided by Gencast and performed a 15-day autoregressive forecast. The results were compared with both the ERA5 ground truth and our own predictions for lead times ranging from 5 to 15 days. The forecast start date was March 29, 2019, and the figure presents visualizations for lead times of 5, 7, 10, and 15 days(from top to bottom). The three columns correspond to ERA5 ground truth (left), ours(center), and Gencast predictions (right).

