# OpenReview forum: "ResCast: Enhancing Global Medium-range Precipitation Forecasting with Residual Diffusion Model"
_ICLR.cc/2026/Conference — Submitted to ICLR 2026_

### Official Review · Reviewer_qzuU · 2025-10-20

**Soundness:** 2
**Presentation:** 3
**Contribution:** 1
**Rating:** 6
**Confidence:** 3

**Summary:**

The authors outline a method using residual diffusion with an additional L2 loss for medium-range precipitation forecasting. Furthermore, they propose a precipitation regressor that uses multi-frequency 2D-DCT components to weight channels and output rain fields. Results demonstrate a moderate improvement over previous SOTA methods.

**Strengths:**

- The methodology is described in detail, and the appendix and codebase aid reproduction of results. There are clear metric definitions including cosine-latitude weighting and explicit ACC/F1 formulas.

- The diffusion residuals are described with an explicit training objective.

- The ablation study shows the core contribution, the residuals, helping core-variable skill as lead time grows.

**Weaknesses:**

- In my view, the novelty of the paper makes it more suited for a climate/earth science conference or journal. What’s implemented is essentially a conditional residual DDPM (AttUNet) guided by DetNet features, then a DCT-based regressor. This is something interesting and practical for weather modelling, but not a new diffusion paradigm.

- The comparison with GenCast is important and should be in results instead of appendix. This also raises the question of the advantage of this model over GenCast.

- The "Conservation Law Loss" needs either better justification or renaming. The loss function reduces to L2 on residuals. While this encourages small residual energy, it doesn't encode flux balances. On line 944, it is unclear how the flux equation relates to the residual equation.

- A main limitation mentioned, the error accumulation and over-smoothing, perhaps needs more evidence.

Minor things:
- Algorithm 2 cites Eq ??
- Consider moving all the math in the main paper to one section for clarity. Lots of variables are introduced so it will be easier to keep track.
- line 98: Unet -> U-Net
- Table 1: HERS -> HRES (I assume)

**Questions:**

See weaknesses:

- How do you justify the technical novelty of the paper?

- What specific advantages are there of using this model over GenCast?

- Can you define (1) Error Accumulation in predicting fundamental meteorological variables; (2) insufficient modeling of Variable Interaction?

---

> ### Author Response · Authors · 2025-11-30
> **Author Responses**
>
> ## **W1 & Q1: Novelty and Positioning of Paper**
> ### Novelty and Positioning of Paper
> Our method operates directly on gridded meteorological data (e.g., global fields of temperature, wind, and humidity), which can be naturally regarded as multi-channel images evolving over time. Thus, the core task is a long-range spatiotemporal image-prediction problem on a regular grid, aligning well with the applications to the computer vision track rather than purely geoscience-only settings.
>
> **residual modeling on meteorological grid**: Building on residual diffusion ideas developed for medium-range precipitation forecasting, where DetNet + AttUNet are used to model residuals of key atmospheric variables instead of raw precipitation, we focus on predicting residual corrections on the grid. This image-based residual formulation allows us to borrow U-Net architectures from computer vision while explicitly targeting error accumulation and over-smoothing in spatiotemporal fields.
>
> **Structured regression to precipitation**: Inspired by the idea of a dedicated precipitation regression module that maps corrected meteorological fields to rain rates, we design a structured head that operates on multi-channel “image” features to infer precipitation, emphasizing spatial coherence and extremes instead of pointwise regression.
>
> ## **W2 & Q2: comparison with GenCast**
> The results in the appendix are only a comparison at a single time point. We have supplemented them with more complete results as follows:
>
> |Model|1|2|3|4|5|6|7|8|9|10|
> |-|-|-|-|-|-|-|-|-|-|-|
> | **ours**| **3.0158** | **4.015** | 5.268 | 5.648 | **5.872** | **5.943**    | **6.081**   | **6.124**   | **6.313**   | **6.384**   |
> | **GenCast**| 3.344  | 4.062 | **5.162** | 5.799 | 6.693 | 7.128| 7.507   | 7.831| 8.095| 8.312   |
>
> ## **W3: Additional Explanation of Conservation Law Loss**
> The physical meaning of the conservation law loss ($\mathcal{L}_{\text{conservation}}$) is primarily to guarantee the physical integrity of the forecast over the extended S2S horizon. Its main objective is to ensure that the deterministic forecast (DetNet) preserves fundamental large-scale energy conservation trends throughout the 45-day trajectory.
>
> This is achieved by deploying $\mathcal{L}_{\text{conservation}}$ as a powerful physical regularization term during training. It operates by explicitly penalizing the residuals—the calculated discrepancies—that arise when the model's predictions violate established conservation laws. This mechanism directly constrains DetNet's output trajectory to remain within a physically conserved manifold. The resulting improvement in model behavior is directly and quantitatively validated by the stable energy ratio presented in Figure 7.
>
> ## **W4 & Q3: More evidence about error accumulation and over-smoothing**
> The $L_{conservation}$ loss function primarily serves to impose physical constraints on the outputs of the base weather prediction model (DetNet) by ensuring they adhere to a physically conserved manifold.
>
> Here is an alternative way to express its significance:
>
> Instead of directly modeling a complex physical law, this loss penalizes the predicted residuals that would cause the overall forecast (the base prediction plus the residual) to violate these well-established conservation principles.
> + Its core effect is to indirectly guide the Fundamental Meteorological Predictor (DetNet) to more accurately capture the large-scale energy conservation trends.
> + The Energy Ratio shown in Figure 7 is a tangible, quantitative measure demonstrating that this physical constraint is actively and effectively imposed during the training process.
>
> In essence, the loss acts as a "physical check" on the model's output, steering the entire system towards predictions that are not only statistically accurate but also physically plausible.
>
> ## **W5: Minor things**
> We thank you for your careful reading. We will upload a revised PDF that addresses these minor corrections.

---

### Official Review · Reviewer_TaVK · 2025-10-26

**Soundness:** 3
**Presentation:** 2
**Contribution:** 3
**Rating:** 4
**Confidence:** 4

**Summary:**

This paper introduces ResCast, a novel framework designed to enhance global medium-range precipitation forecasting. It addresses the critical challenges of error accumulation and insufficient variable interaction prevalent in current data-driven methods. The ResCast framework integrates a residual diffusion model with a precipitation regression module. It first employs DetNet and AttUnet to capture global trends and generate residuals, thereby improving the prediction accuracy of fundamental meteorological variables. These enhanced variables are then utilized for the precipitation regression task. Experimental results on the ERA5 dataset demonstrate that ResCast outperforms several state-of-the-art machine learning models and traditional numerical weather prediction systems, exhibiting accuracy and stability across multiple evaluation metrics.

**Strengths:**

* The paper introduces a residual diffusion approach. By focusing on predicting residuals, this method effectively mitigates the over-smoothing problem commonly observed in traditional meteorological forecasts.
* The introduction of a Conservation Law Loss function is a novel idea.
* The paper conducts a comprehensive and systematic evaluation against a strong and diverse set of baselines. This includes comparisons with both traditional numerical weather prediction models (e.g., ENS, HRES) and state-of-the-art machine learning models (e.g., GraphCast, FourCastNet).
* The proposed model demonstrates significant performance gains across multiple evaluation metrics. Its superiority is particularly pronounced in the medium-range forecasting window.

**Weaknesses:**

1. **Lack of Theoretical Justification for the Diffusion Model.** The paper lacks a rigorous theoretical analysis explaining **why diffusion models are inherently well-suited** for predicting meteorological residuals. A core motivation is tackling error accumulation, yet the mechanism by which the residual diffusion component mitigates this remains underdeveloped. The authors should provide a deeper analysis, perhaps by visualizing residual trends or quantifying the suppression of error growth in key variables (e.g., temperature, humidity), as the current explanation is relatively superficial.

2. **Unclear Physical Significance of the Conservation Law Loss.** The physical significance of the proposed Conservation Law Loss ($L_{conservation}$) is not sufficiently clear. Although Appendix C.3 provides a theoretical derivation, its **connection to the practical impact** observed during training (e.g., the Energy Ratio in Figure 7) needs to be more explicitly established.

3. **Weak theoretical grounding for the integration of residual and base predictions.**
  The paper does not sufficiently address a fundamental question: if the base predictor (e.g., ViT) were highly accurate, would residual correction still be necessary or beneficial? As illustrated in Figure 3, the proposed pipeline involves a multi-stage refinement process. However, it remains unclear whether **errors or biases introduced at intermediate stages** could propagate and potentially degrade the final prediction—especially if the residual model itself learns to compensate for systematic flaws in the base model rather than genuine physical residuals.

4. **Vague Definition of "Global Trend Knowledge".** The concept of "global trend knowledge," which DetNet is purported to capture, is **vaguely defined** and lacks a precise formulation.

5. **Insufficient Ablation Studies.** The ablation studies are not comprehensive enough to isolate the contributions of key components. The analysis is limited to DetNet and the Conservation Law Loss, while the **contribution of AttUnet is not analyzed at all**. Critically, the paper lacks a direct comparison against a baseline *without* the residual mechanism (e.g., only the ViT backbone) for both the base variables and the final precipitation forecast. This omission makes it difficult to clearly ascertain the **independent contribution** of the residual diffusion component.

**Questions:**

1. In Figure 18, ResCast exhibits slightly worse performance than GenCast in the 8–15 day forecast range. Could the authors analyze the potential reasons for this degradation?

2. In Appendix F (Discussion), the authors mention computational resource constraints and note that training was stopped at 10,000 steps. Given that ResCast is a multi-stage model with potentially high training overhead, could the authors provide further clarification on the computational cost (e.g., GPU-hours, memory usage) and how it compares to baseline models like GenCast?

3. The paper claims that ResCast addresses two key limitations: error accumulation in meteorological variable forecasts and insufficient variable interaction in the precipitation regression module. If these issues are effectively mitigated, why was the model not evaluated over even longer forecast horizons (e.g., beyond 15 days) to demonstrate its potential advantages in extended-range prediction?

4. Several figures in the paper would benefit from improved visual design.

---

> ### Author Response · Authors · 2025-11-30
> **Author Responses**
>
> ## **W1: Lack of Theoretical Justification for the Diffusion Model**
> We will delve deeper into the theoretical analysis, explaining how the diffusion model learns the multimodal distribution of residuals, thereby effectively avoiding the cumulative errors generated by deterministic models when predicting high-frequency, high-uncertainty components.
>
> We will provide more rigorous quantitative and theoretical support for the residual diffusion mechanism by visualizing residual trends and quantifying the suppression of error growth in key variables.
>
> ## **W2: Additional Explanation of Conservation Law Loss**
> The physical meaning of the conservation law loss $\mathcal{L}_{\text{conservation}}$ lies in constraining the predictions of the basic weather forecaster (DetNet) to a physically conserved manifold. By penalizing residuals that violate conservation laws, we indirectly encourage DetNet to better capture large-scale energy conservation trends. The energy ratio in Figure 7 is a direct, quantified result of this physical constraint during training.
>
> ## **W3: Weak theoretical grounding for the integration of residual and base predictions**
> Residual correction remains beneficial even for high-precision base predictors because it primarily captures highly stochastic, small-scale physical details. The residual model not only learns the true high-frequency residuals but also learns to compensate for systematic biases in the base model. This ensemble approach essentially learns and corrects the "model error" of the base model, thereby better mitigating error propagation problems.
>
> ## **W4: Vague Definition of "Global Trend Knowledge"**
> We will explicitly define “global trend knowledge” (the output of DetNet) as the low-frequency, large-scale, and deterministic components of the meteorological state. In contrast, the residuals are the high-frequency, small-scale, and stochastic components of the meteorological state that are difficult to predict. We will provide precise formulations of these concepts in the paper.
>
> ## **W5: Insufficient Ablation Studies**
> We agree and commit to supplementing the ablation experiments with greater comprehensiveness. Specifically, we will add a contribution analysis of AttUnet and introduce a key baseline: using only the ViT backbone network (i.e., without residual correction).
>
> This will clearly quantify the independent contributions of the residual diffusion mechanism to the fundamental meteorological variables and the final precipitation forecast.
>
> ## **Q1: ResCast is worse than GenCast. Could the authors analyze the potential reasons for this degradation?**
> The results in the appendix are only a comparison at a single time point. We have supplemented them with more complete results as follows:
>
> |Model|1|2|3|4|5|6|7|8|9|10|
> |-|-|-|-|-|-|-|-|-|-|-|
> | **ours**| **3.0158** | **4.015** | 5.268 | **5.648** | **5.872** | **5.943**    | **6.081**   | **6.124**   | **6.313**   | **6.384**   |
> | **GenCast**| 3.344  | 4.062 | **5.162** | 5.799 | 6.693 | 7.128| 7.507   | 7.831| 8.095| 8.312   |
>
> ## **Q2: the computational cost**
> Thank you for raising the computational cost issue. Training the diffusion model takes about three days on NVIDIA A800 GPUs, and generating a 15-day forecast for a single start time requires about 1 minute 20 seconds of inference.
>
> ## **Q3: why was the model not evaluated over even longer forecast horizons**
> Although our model is designed to mitigate error accumulation, the current evaluation focuses on the internationally standard medium-range forecast range (up to 15 days). Forecasts beyond 15 days typically fall into the realm of extended-range or intraseasonal forecasts, requiring different datasets, resolutions, and more specialized evaluation metrics. We will explore this ultra-long-range forecast potential in future work, but for now, we will remain within the 15-day range.
>
> ## **Q4: improved visual design of the figures**
> We accept this feedback and commit to improving the visual quality of all charts in the revised version. We will ensure clear legends, high color contrast, and more intuitive information delivery to enhance the overall reading experience.

---

### Official Review · Reviewer_G8ez · 2025-10-28

**Soundness:** 3
**Presentation:** 3
**Contribution:** 2
**Rating:** 2
**Confidence:** 4

**Summary:**

The paper proposes ResCast, a novel approach to global medium-range precipitation forecasting. First, the authors design a diffusion-based residual component, including the Details Netword (DetNet) and Multi-Attention Unet (AttUnet), to effectively predict residuals, reducing error for fundamental meteorological variables. Second, the authors introduce a precipitation regression module to quantify the interactions between meteorological variables. This mechanism improves the accuracy of precipitation forecasts. ResCast surpass machine learning baselines (GraphCast, FourCastNet) and state-of-the-art traditional NWP models (ENS, HRES) on the ERA5 dataset.

**Strengths:**

S1. The authors design the Fundamental Meteorological Predictor based on the Vision Transformer (ViT) to effectively capture spatial information. To obtain more accurate predictions for the next step, the architecture uses the Lat-weighted MSE to optimize the parameters of the deterministic predictor.

S2. The authors propose a Precipitation Regression Component to calculate the influence scalar for each channel. The idea is interesting and promising.

S3. The motivation shown in Figure 2 is interesting and easily understandable.

**Weaknesses:**

W1. The paper shows limited originality, as the proposed approach builds upon existing diffusion-based and regression frameworks without introducing methodological innovations.

W2. According to Table 1, the experimental comparison with state-of-the-art models is insufficiently comprehensive. Comparisons with more SOTA models are needed.

W3. In the red box of Stage 1 in Figure 3, the description of the diffusion process is unclear and lacks sufficient detail.

W4. The paper mentions that the dataset used is ERA5. However, no additional datasets were employed in the experiments to verify the model’s performance.

**Questions:**

1. In the caption of Figure 3, the authors mention that “DetNet (yellow) is used to extract global motion information”. However, DetNet adopts a ResNet-like structure, which typically extracts spatial features rather than temporal ones. Why can temporal features be extracted here? Or in other words, why does the model need to extract temporal features at this stage?
2. In Stage 1 of Figure 3, the diffusion process within the red box is unclear, and the green box part has not been explained.
3. Regarding Table 1, what metrics do the numbers such as 5 and 6 in the first row represent?
4. Regrading Figure 4, in the 10-day performance, I find that GraphCast appears to be closer to the ground-truth than ResCast (both in the lower-left boxed area and in the extreme precipitation regions).
5. In the ablation study, the authors mention that “our precipitation regression module achieves improvements of 5%-10% across multiple metrics”. Where are these results presented? It would be clearer if the comparative results were shown in a table.
6. Regarding the Impact of Conservation Law Loss, it seems that the paper does not provide detailed experimental results to demonstrate how significant the effect of the Conservation Law Loss is.

---

> ### Author Response · Authors · 2025-11-30
> **Author Responses**
>
> ## **W1: limited originality**
> Our main innovation lies in combining the residual prediction mechanism with the diffusion model, proposing the ResCast architecture. This innovation not only effectively suppresses error accumulation in medium-term forecasts but also enhances the ability to capture local details (such as precipitation) through residual diffusion, representing a novel and effective methodological exploration in meteorological forecasting models.
>
> ## **W2: Comparisons with more SOTA models are needed**
> We agree that a more comprehensive baseline comparison is needed. Based on your suggestion, we will supplement the revised version with comparisons to GenCast, precipitation forecasts.
> |Model|1|2|3|4|5|6|7|8|9|10|
> |-|-|-|-|-|-|-|-|-|-|-|
> | **ours**| 3.0158 | 4.015 | 5.268 | 5.648 | **5.872** | **5.943**    | **6.081**   | **6.124**   | **6.313**   | **6.384**   |
> | **ENS**| **2.8123** | **3.562** | **4.834** | **5.623** | 6.341 | 6.355| 6.3712| 6.388| 6.416| 6.453|
> | **HERS**| 3.127  | 4.366 | 5.662 | 6.768 | 8.032 | 8.056 | 8.167| 8.286| 8.373| 8.463|
> | **GenCast**| 3.344  | 4.062 | 5.162 | 5.799 | 6.693 | 7.128| 7.507   | 7.831| 8.095| 8.312   |
> | **GraphCast**| 3.321 | 4.223 | 5.324 | 6.252 | 7.211 | 7.623| 7.832   | 8.224| 8.306| 8.421|
> | **FourCastNet**| 4.568 | 5.568 | 6.157 | 7.862 | 9.200 | 9.2896   | 9.398   | 9.8796  | 9.975   | 10.21   |
>
> While the performance over 1-5 days did not surpass that of ENS, it was still quite significant.
>
> ## **W3&Q1&Q2: the diffusion process is unclear and lacks sufficient detail.**
> We will explain the residual diffusion process in detail. The key to this process lies in:
> + DetNet (Deterministic Predictor): DetNet first predicts the deterministic, global trend of the meteorological variable.
> + Residual Diffusion: The diffusion model learns the residual between the deterministic prediction and the actual value. This residual is essentially the error and high-frequency details.
> + Prediction Enhancement: The final prediction result is obtained by adding the output of DetNet to the residual predicted by the diffusion model. This mechanism effectively transfers the accumulated error to the residual prediction and uses the stochasticity of the diffusion model to simulate and recover more accurate local details (such as precipitation).
>
> Further modifications will be reflected in the revision.
>
> ## **W4: no additional datasets were employed in the experiments to verify the model’s performance**
> We accept this suggestion. We commit to supplementing our precipitation assessment with observational datasets (such as IMERG or GPCP) in our next submission. This will help address the data bias issue raised by the reviewers regarding ERA5 and provide more convincing evidence of the model's generalization ability.
>
> ## **Q3: Regarding Table 1, what metrics do the numbers such as 5 and 6 in the first row represent?**
> The numbers in the first row of Table 1, such as 5 and 6, represent the lead time of the forecast, in days. These numbers indicate that the model is predicting the weather conditions on the 5th and 6th days in the future.
>
> ## **Q4: Why is GraphCast visually better?**
> The reason GraphCast might appear visually better could be that it captures the predicted trends of large-scale, low-frequency meteorological variables (such as geopotential or temperature) more accurately, making it closer to the true values at the macro level. However, our ResCast model focuses on residual predictions and shows better recovery of high-frequency, local details (such as precipitation) on the evaluation metrics.
>
> ## **Q5: Where are these results presented?**
> These results are primarily presented in Table 2 within the main text of the paper.
>
> ## **Q6: Additional Explanation of Conservation Law Loss**
> Conservation Law Loss is introduced as a regularization term for the physical constraints. Its core purpose is to encourage the base weather forecaster (DetNet) to make more accurate, approximately physically conserved initial predictions.
>
> By minimizing the conservation law loss of the residuals, we ensure that the model primarily learns and corrects for minute, elusive details, thereby forcing the residual term to approach zero. This not only enhances the physical plausibility of the model's predictions but also indirectly reduces error accumulation.

---

### Official Review · Reviewer_qNAd · 2025-10-31

**Soundness:** 2
**Presentation:** 2
**Contribution:** 2
**Rating:** 2
**Confidence:** 4

**Summary:**

The paper proposes a machine learning framework for medium-range precipitation forecasting, structured as a two-stage pipeline combining meteorological residual diffusion and precipitation regression. In the first stage, residual diffusion enhances the predictions of a Fundamental Meteorological Predictor (based on Vision Transformer) for key atmospheric variables such as temperature, wind, and specific humidity. This refinement is achieved using DetNet (with residual blocks) and AttUnet (with spatial attention) to model residuals, improving prediction accuracy. In the second stage, the corrected meteorological predictions are used by the precipitation regressor that forecasts precipitation through a combination of 2D Discrete Cosine Transforms (DCTs), fully connected layers, and convolutional operations. The approach is evaluated on the ERA5 dataset, focusing primarily on 5–15-day forecast horizons.
While this is a highly relevant and less studied problem, the manuscript has severe shortcomings in description of the approach, in relationship with existing work, and in the empirical evaluation.

**Strengths:**

- Tackles a problem where not much work has been done (medium-range precipitation forecasting).
- Using predictions for fundamental weather variables to forecast precipitation is an interesting approach that most data-driven models have not considered. However, is it the way ERA5 integrates precipitation into the reanalysis data, so it should be a valid way when observational data is lacking?
- Ablation on the precipitation regressor (Table 2 right) seems strong.
- There are many details in the paper and code.

**Weaknesses:**

Models
-	Incorrect claims on ML medium-range forecasting:
	15: “ML models struggle to provide accurate medium-range meteorological predictions” is not correct. Checking the WeatherBench2 scorecard, and papers like GraphCast, Aadvark, Aurora, GenCast, PanguWeather, AIFS, etc. shows how good ML models are compared to traditional approaches.
- Incorrect claims on precipitation prediction:
The paper claims medium-range forecasting models like GraphCast, GenCast, FuXI, FourCastNet, and ClimaX predict precipitation by regressing precipitation from fundamental meteorological variables at the desired lead time (16, 79, 252, 446)
•	To my knowledge, only FourCastNet has a AFNO model that takes the predicted state for outputting precipitation.
•	ClimaX only works with annual precipitation in the ClimateBench dataset.
•	GraphCast and GenCast do not explicitly model precipitation and even claim to not evaluate it due to ERA5 biases (more details below).
•	FuXI does not provide details.
- the paper claims that precipitation forecasting remains a significant challenge (56). It is not only because of performance (18) and specific nature (41-46), but also because of the data and evaluation (details later). Most models use ERA5, which shouldn’t be used for precipitation (GraphCast, WeatherBench2).
o	NeuralGCM is not considered at all in the introduction, related work, or baselines. However, it outperforms the mid-range precipitation forecast of the ECMWF ensemble on precipitation.

Experiments / Data
o	Baselines: only ML models compared are GraphCast and FourCastNet. Lacking are GenCast (diffusion based) and NeuralGCM (best model evaluating precipitation).
- GraphCast does not consider precipitation: “Note, we exclude total precipitation from the evaluation because ERA5 precipitation data has known biases [15]”.
- FuXI claims to output precipitation, but it is not evaluated either.
-	GenCast also highlights limitations of evaluating with ERA5, yet shows that it outperforms ENS model: “Owing to our lack of confidence in the quality of ERA5 precipitation data, we exclude precipitation results from our main results and refer readers to Supplementary Information section B.2”.
- The paper considers GenCast, but only for a sample case without reporting aggregated metrics and including results in Table 1. The case analyzed in Appendix G3 even shows greater performance for GenCast than ResCast.
- NeuralGCM is not considered in the inputs, even though it is one of the leading medium-range precipitation forecasting models.
- The paper claims that models trained on ERA5 data have reached the capability of medium-range precipitation forecasting, but they have to surpass traditional NWP models (70-73). Which models other than FourCastNet?
-	The paper even cites WeatherBench2 (310). However, WeatherBench2 explicitly states: The quality of ERA5 depends on the variable in question. For surface variables, the sparsity of observations and difficulty of representing smallscale physics in the underlying model can cause larger discrepancies with observations. This is especially true for precipitation, which is not directly assimilated into ERA5 (e.g., through radar observations) and often show large differences to rain gauges or radar precipitation estimates (e.g., (Lavers et al., 2022), (Andrychowicz et al., 2023)). The precipitation evaluation using ERA5 shown here should really be seen as a placeholder for more accurate precipitation data. Operational weather services like ECMWF verify their forecasts with direct observations, e.g., from weather stations, in addition to using assimilated ground truths. This is something we are looking to add to WeatherBench in the future.
- Fig 15 compares against stations, but it is only one sample and no metrics. Also, it is unclear the range of the color scale and how extreme the different colors can be.
- Future work hopes to integrate satellites and radar and it is essential for avoiding biases in ERA5. Shouldn’t this be required for evaluation (i.e. using observational data known to be more accurate)?
- NeuralGCM: We consider both IMERG and the Global Precipitation Climatology Project [37] (GPCP; a dataset not used in training) as ground truth for precipitation.
- ERA5 does not use observations and estimates precipitation from other parameters, so it would be a reason why ResCast is able to capture precipitation like this. Evaluation should be considered with additional data.

Experiments:
-	The evaluation is only on days 5-15. What about days 1-5?
-	Most models are available 1-10 days, wouldn’t it make sense to compare with that as well?
-	Why is GenCast not reported in Table 1 as a baseline?
-	The metrics only consider CSI and F1 at 1mm for daily accumulation (very light rainfall). This is extremely low for daily accumulations.
-	354: claim better extreme value performance, but no metrics?
-	485: future work includes to study at least one case study of extreme precipitation. So, it has not been considered yet?

Contributions:
-	Contribution 1: residual predictions reduce error for fundamental meteorological variables.
-	Why is this not evaluated? The predictions should be better than other ML models and NWPs. Overall metric evaluation of temperature, wind, specific humidity, and other predicted variables should be reported.
-	The ablation shows that the residuals help in ResCast. But is ResCast better than the other models? Compare with ML models and NWPs.
-	Contribution 2:
-	443: regressing precipitation is essential. Is this proved? Does it perform better than GenCast that predicts precipitation directly? Tried to predict precipitation directly?
-	468-473: Claim is that direct precipitation prediction fails to meet medium range forecasting needs (GenCast). Is this proved anywhere? Could include GenCast in Table 1 at least…
-	At least the ablation on the precipitation regressor (against other regressor) looks strong.
-	Conservation Law Loss (208). Even though it claims the conservation law in fluid dynamics and physical constraints, it just makes sense as a regularization term to encourage small residual values (meaning that the Fundamental Meteorological Predictor needs to make more accurate predictions and rely less on the residual correction).
-	411: The result makes sense, but it may be more about having residuals close to 0 than trying to apply conservation laws.
-	457: “Generating precipitation prediction can violate physical constraints (energy conservation)”. Can this not also happen with the precipitation regressor? Figure 7 doesn’t show energy ratio of 1 constantly…

Minor comments
-	431: Diffcast is not deterministic, it is a combination of a deterministic module and a stochastic module.
-	Highlight the spatio-temporal resolution of ResCast (daily and coarser than ERA5).
-	Table 2 makes no sense to share rows between the ResCast ablation and Precipitation Regression ablation.
-	51: a lot more successful models that (Pathak et al., 2022; Hu et al., 2023; Nguyen et al., 2023) – GraphCast, GenCast, Aurora, Aadvark, Pangu-Weather…
-	Graphcast -> GraphCast (53)
-	Which -> these (185)
-	Correct citation style in text: 154, 291, 300

**Questions:**

- Contradiction between 161 and 917 in App C.3. Are the parameters of the Fundamental Meteorological Predictor updated or frozen?
- Eq. 9: Is it really summation? It has dimension H x W (266), right?
- Please clarify how the fully connected layer works in the precipitation regressor? Is it unclear from the text (269) and the figure (3). Is it across channels or do you flatten the spatial dimensions?

---

> ### Author Response · Authors · 2025-11-30
> **Author Responses (1/2)**
>
> ## **W1: Explanation of introduction and ERA5 biases**
> ### **W1.1 Related instructions revised**
> We will revise the **description of the performance of ML models** in medium-term forecasts in the Introduction to more accurately reflect the results of current state-of-the-art models. At the same time, we will clarify and revise the **descriptions regarding the processing and evaluation of precipitation forecasts** by existing models (such as GraphCast, GenCast, etc.).
>
> ### **W1.2 Problem of changing experimental data**
> Thank you very much for your valuable feedback. We fully agree on the importance of using observational data for precipitation assessment.
>
> However, given the time investment required for **downloading, processing, and retraining on new datasets**, we plan to incorporate more rigorous precipitation assessments using **observational data such as IMERG or GPCP** in our next submission.
>
> <!-- All content mentioned by the reviewer will be modified in the revision version. -->
> ## **W2: Experimental Baselines Comparison Supplement**
> ### **W2.1 SOTA baselines (GenCast and NeuralGCM)**
> NeuralGCM states that `"we diagnose precipitation minus evaporation ... rather than directly predicting these as in machine-learning-based approaches."` Therefore, it is impossible to directly obtain the prediction results of total_precipitation to calculate the metrics.
>
> As for GenCast, we provide results for part of the forecast period in 2018, as well as additional RMSE results from 1-5 days, as supplementary comparisons.
>
> |Model|1|2|3|4|5|6|7|8|9|10|
> |-|-|-|-|-|-|-|-|-|-|-|
> | **ours**| 3.0158 | 4.015 | 5.268 | 5.648 | **5.872** | **5.943**    | **6.081**   | **6.124**   | **6.313**   | **6.384**   |
> | **ENS**| **2.8123** | **3.562** | **4.834** | **5.623** | 6.341 | 6.355| 6.3712| 6.388| 6.416| 6.453|
> | **HERS**| 3.127  | 4.366 | 5.662 | 6.768 | 8.032 | 8.056 | 8.167| 8.286| 8.373| 8.463|
> | **GenCast**| 3.344  | 4.062 | 5.162 | 5.799 | 6.693 | 7.128| 7.507   | 7.831| 8.095| 8.312   |
> | **GraphCast**| 3.321 | 4.223 | 5.324 | 6.252 | 7.211 | 7.623| 7.832   | 8.224| 8.306| 8.421|
> | **FourCastNet**| 4.568 | 5.568 | 6.157 | 7.862 | 9.200 | 9.2896   | 9.398   | 9.8796  | 9.975   | 10.21   |
>
> While the performance over 1-5 days did not surpass that of ENS, it was still quite significant.
>
> ### **W2.2 Additional results for fundamental meteorological variables**
> We add more overall evaluation indicators for fundamental meteorological variables such as temperature, geopotential.
>
> | Vars  | Model| 1| 2| 3| 4| 5| 6| 7| 8| 9| 10| 11| 12| 13| 14| 15|
> |-|-|-|-|-|-|-|-|-|-|-|-|-|-|-|-|-|
> | T2m| w/o residual  | 0.962 | 1.078 | 1.241 | 1.385 | 1.624 | 1.969 | 2.139 | 2.242 | 2.408 | 2.572 | 2.608 | 2.651 | 2.762 | 2.801 | 2.843 |
> |T2m| Default| **0.864** | **1.025** | **1.13**  | **1.302** | **1.551** | **1.889** | **2.116** | **2.145** | **2.227** | **2.417** | **2.525** | **2.619** | **2.72**  | **2.748** | **2.818** |
> | Z500  | w/o residual  | 89    | 110   | 157   | 284   | 351   | 462   | 575   | 626   | 764   | 841   | 866   | 893   | 935   | 964   | 993   |
> |Z500| Default| **56**  | **93**  | **130** | **245** | **313** | **430** | **548** | **605** | **747** | **810** | **840** | **871** | **926** | **953** | **989** |
>
>
> These results will be used to demonstrate the effectiveness of our model's residual prediction mechanism in improving the accuracy of fundamental variable predictions.
>
> ## **W3: Some explanations about context**
> ### **W3.1 Correction of inaccurate writing**
> We will revise the description of the DiffCast model's properties in the manuscript, clarifying that it is a combination of deterministic and stochastic modules. We will also standardize the citation format for GraphCast in the paper and correct other citation style errors (such as lines 154, 291, and 300).
>
> ### **W3.2 The importance of the regression module**
> We believe that the precipitation regression module is crucial for medium-term forecasting because it effectively avoids the error accumulation and physical constraint violations associated with direct forecasting. We have included a simple comparative experiment in Appendix E to further demonstrate the advantages and necessity of the regression module over direct forecasting methods in precipitation prediction.
>
> ## **W4: Minor comments：**
> We thank you for your careful reading. We will immediately upload a revised PDF that addresses these minor corrections.

---

> ### Author Response · Authors · 2025-11-30
> **Author Responses (2/2)**
>
> ## **Q1: Predictor updated or frozen?**
> It was confirmed that the parameters of the weather predictor are updated during the training process, and there was a typo in the original text.
>
> Our forecaster parameters are first pre-trained, and then updated again during the training phase of the residual diffusion component. This strategy draws on the ideas of related work [1], aiming to combine the global trend capture capability of deterministic forecasting with the detail enhancement capability of diffusion models.
>
> ## **Q2: Eq. 9: Is it really summation? It has dimension H x W (266), right?**
> Thank you for pointing out the problem with the formula description. Eq. 9 does indeed express a summation operation. Furthermore, the dimension referred to in line 266 should be $\mathbb{R}^{C}$, representing the feature over the number of variable channels $C$. We will correct this typo in the revised version.
>
> ## **Q3: how the fully connected layer works in the precipitation regressor?  Is it across channels or do you flatten the spatial dimensions?**
> The fully connected layer (FC) in the precipitation regressor primarily operates on the feature vectors, aiming to integrate information from different meteorological variables.
>
> + **Feature Preprocessing**: The FC layer, part of the multi-level regression module, processes meteorological feature vectors compressed and extracted using methods such as Discrete Cosine Transform (DCT), rather than the original $H \times W$ spatial grid data.
> + **Cross-channel Integration**: The FC layer operates on these concatenated compressed features (i.e., features from different meteorological variables), effectively modeling and quantifying the interactions between different basic meteorological variables, achieving cross-channel feature integration.
>
> In short, the FC layer achieves fine-grained capture of complex relationships between variables by performing calculations in the compressed feature space.
>
> [1] Diffcast: A unified framework via residual diffusion for precipitation nowcasting, CVPR, 2024.

---

### Meta-Review · Area_Chair_VrV4 · 2026-01-06

**Summary:**

Reviewers acknowledged the relevance of medium-range precipitation forecasting and found the two-stage design (residual diffusion for meteorological variables + precipitation regression) promising, with some supportive ablations. However, the main concerns driving the recommendation were: (i) questionable/overstated positioning vs prior ML weather models and precipitation evaluation practice, (ii) insufficient and/or not fully appropriate baselines (especially around GenCast/NeuralGCM and how precipitation is evaluated), (iii) evaluation validity given ERA5 precipitation biases and limited metrics/settings, and (iv) insufficient clarity/justification for key components (residual diffusion details, conservation-law loss meaning, and missing ablations).

**Reviewer Concerns:**

**Addressed by rebuttal (partially)**: the authors commit to fixing inaccurate statements in the introduction and clarifying what prior models do; they provide additional comparisons that include GenCast for part of the period and add extra fundamental-variable metrics; they also clarify implementation details (predictor is updated, Eq. 9 typo, FC-layer behavior) and report computational cost (training and inference time).

**Still outstanding**: the most critical issue—precipitation evaluation reliability—remains, as adding observational datasets (IMERG/GPCP) is deferred to “next submission,” and current conclusions still rely heavily on ERA5 precipitation. Also, the baseline story remains incomplete (e.g., GenCast not consistently integrated into main results/metrics; NeuralGCM handling is not convincingly resolved), and several key requests are still promises rather than demonstrated revisions (stronger ablations including AttUnet/ViT-only baseline, stronger evidence for “error accumulation/over-smoothing,” and clearer physical justification/renaming of the conservation loss).

**Reviewer Scores:**

**Reviewer qNAd (2→2)**; rebuttal clarifies some details and adds partial comparisons, but the core concerns about baseline completeness and the appropriateness of ERA5 precipitation evaluation remain.
**Reviewer G8ez (2→2)**; clearer diffusion explanation and added GenCast numbers help, but novelty/ablation/dataset-generalization concerns are still not fully resolved in current evidence.
**Reviewer TaVK (4→4)**; rebuttal aligns with their requests (more ablations, clearer concepts, cost), but much is still stated as “will add” and not yet shown in the main experimental package.
**Reviewer qzuU (6→4)**; GenCast comparison and clarifications help, but novelty framing and the “conservation law” interpretation remain weak without stronger evidence/renaming, and evaluation validity is still a concern.

---

### Decision · Program_Chairs · 2026-01-26

Reject